# LoPRo: Enhancing Low-Rank Quantization via Permuted Block-Wise Rotation

## Abstract

Post-training quantization (PTQ) enables effective model compression while preserving relatively high accuracy. Current weight-only PTQ methods primarily focus on the challenging sub-3-bit regime, where approaches often suffer significant accuracy degradation, typically requiring fine-tuning to achieve competitive performance. In this work, we revisit the fundamental characteristics of weight quantization and analyze the challenges in quantizing the residual matrix under low-rank approximation. We propose LoPRo, a novel fine-tuning-free PTQ algorithm that enhances residual matrix quantization by applying block-wise permutation and Walsh-Hadamard transformations to rotate columns of similar importance, while explicitly preserving the quantization accuracy of the most salient column blocks. Furthermore, we introduce a mixed-precision fast low-rank decomposition based on rank-1 sketch (R1SVD) to further minimize quantization costs. Experiments demonstrate that LoPRo outperforms existing fine-tuning-free PTQ methods at both 2-bit and 3-bit quantization, achieving accuracy comparable to fine-tuning baselines. Specifically, LoPRo achieves state-of-the-art quantization accuracy on LLaMA-2 and LLaMA-3 series models while delivering up to a $4\times$ speedup. In the MoE model Mixtral-8x7B, LoPRo completes quantization within 2.5 hours, simultaneously reducing perplexity by 0.4↓ and improving accuracy by 8%↑. Moreover, compared to other low-rank quantization methods, LoPRo achieves superior accuracy with a significantly lower rank, while maintaining high inference efficiency and minimal additional latency. The code is available at: https://anonymous.4open.science/r/LoPRo-8C83

## 1 Introduction

Large-scale language models (LLMs) have achieved remarkable success across a wide range of tasks, including text processing and generation. However, as the scope of problems addressed by LLMs expands, these models have grown significantly in size, featuring increasingly complex architectures and parameter counts reaching into the hundreds of millions. To make models suitable for deployment on diverse devices, researchers have explored methods such as pruning, quantization, knowledge distillation, and their combinations. Studies show that quantization generally outperforms pruning in multiple network layers Kuzmin et al. (2023).

In recent years, Post-Training Quantization (PTQ) has received considerable attention for quantizing models without requiring full model retraining (Ding et al., 2022; Hubara et al., 2021; Frantar et al., 2023). An important advancement in this area is the emergence of low-rank compensation (LoRC) PTQ methods (Dettmers et al., 2023; Zhang et al., 2024b), which keep the original model weights unchanged during compensation. This design enables lightweight, low-rank modules to be loaded dynamically when needed, enhancing flexibility and efficiency. (Kwon et al., 2023; Zheng et al., 2024). However, such convenience comes at the cost of performance. Existing approaches Zhang et al. (2024a); Li et al. (2024) often perform poorly under low-bit conditions or require task-specific fine-tuning Zhang et al. (2024a) to achieve acceptable accuracy, limiting their applicability for rapid task adaptation. This motivates our central question: *Is it possible to design a fine-tuning-free low-rank PTQ method that simultaneously achieves optimal quantization accuracy?*

**Challenges 1. Unleashing the potential of low-rank quantization.** Existing low-rank PTQ methods such as SVD-Quant Li et al. (2024) and LQER Zhang et al. (2024a) suffer substantial accuracy

degradation in low-bit settings. Moreover, after the low-rank approximation, residual quantization cannot simply adopt techniques such as rotation, as this undermines the effectiveness of low-rank decomposition.

**Challenges 2. Controlling memory overhead and quantization costs.** The memory costs introduced by low-rank quantization are highly sensitive to rank selection, and performing Singular Value Decomposition (SVD) on large matrices introduces substantial inference latency (for example, using **256 ranks reduces throughput by 45%! Saha et al. (2024)**). The challenge is thus to efficiently achieve high quantization accuracy with a minimal rank.

In summary, this work makes the following contributions:

- We present a comprehensive analysis of existing quantization methods, highlighting the challenges and limitations of low-rank PTO approaches. Based on which, we propose LoPRo — a novel fine-tuning-free low-rank PTQ algorithm based on partial block rotation under permutation. Our method overcomes the incompatibility between low-rank decomposition and other quantization techniques, achieving high accuracy with near 2% extra memory usage and less than 10% latency overhead.

- We introduce a rank-1 mixed-precision refinement of RSVD that halves the storage overhead while maintaining high computational efficiency. On a single GPU, quantizing a 7B model takes less than 0.5 hours, and an 8x7B model can be processed within 3 hours.

- Extensive experiments show that our method delivers state-of-the-art quantization accuracy while also preserving the option for fine-tuning when necessary. Notably, under 3-bit scalar quantization, LoPRo achieves performance that even surpasses the fine-tuned results and maintains good scalability on Mixture-of-Experts (MoE) model.

## 2 RELATED WORKS

In weight quantization, denote the original weight as $W \in \mathbb{R}^{m \times n}$ and compressed into $\hat{W}$; $X \in \mathbb{R}^{n*k}$ is an input from a calibration set; the primary challenge lies in low-bit quantization at 3-bit and below. For PTQs, we summarize three key points that underlie their effectiveness. To align with the experimental setup, we use distinct colors ▮▮▮ bar to represent these methods, where the three color segments correspond to the fine-grained strategies (e.g., a, b, c) within ¶O1, ¶F1, ¶F2 and corresponding to the Tags in §4:

To align with the experimental setup, we employed a colorbar

***Optimization 1 (O1):*** $\|WX - \hat{W}X\|$. Optimizing the output of linear layers directly—i.e., minimizing the loss between full-precision and quantized outputs is the most straightforward way.

***Feature 1 (F1): Weight distribution.*** The weight featuring low rank Hu et al. (2022) and sub-Gaussian distribution Narkhede et al. (2022) contains a few large absolute outliers amidst small numbers.

***Feature 2 (F2): Important channel in activation.*** According to research Lin et al. (2024b), a small subset of high-magnitude activations plays a crucial role in quantization performance.

To address these three points, researchers have proposed a series of algorithms.

**Solution to O1▮:** **Minimizing loss.** Aims to minimize proxy loss by optimizing Nagel et al. (2020):

$$\mathcal{L}(W) = E_X\left[\|WX - \hat{W}X\|\right] = \text{tr}\left((\hat{W} - W)H(\hat{W} - W)^\top\right), \tag{1}$$

where $H = E_X[XX^T]$ is a proxy Hessian. The implementation of this principle follows two paths:

**(a)** **Optimization in quantization:** GPTQ Frantar et al. (2023) quantizes weight columns sequentially and compensates for the loss of each column in subsequent columns, GPTAQ Li et al. (2025) extends this framework to asymmetric calibration. OminiQuant Shao et al. (2023) applies the loss in weight clipping. MoEQuant Chen et al. (2025) provides a better calibrate-set selection to balance the loss on experts.

**(b)** **Optimization through fine-tuning:** This aspect typically performs fine-tuning by optimizing the loss function $\mathcal{L}$ in Eq 1. Representative approaches include fine-tuning the codebook,

as in QUIP# Tseng et al. (2024), and LoRA adaptation such as RILQ Lee et al. (2025), QERA Zhang et al. (2024b), and CALDERA Saha et al. (2024).

**Solution to F1■: Adapting to the Distribution.** Adjust input/output according to the distribution to facilitate quantization:

(a) **Outlier Truncation:** To mitigate quantization loss by outliers, OminiQuant Shao et al. (2023) introduces Learnable Weight Clipping (LWC), which optimizes asymmetric clipping thresholds by minimizing Eq 1.

(b) **Smoothing by Rotation:** Orthogonal transformations are employed to redistribute weight magnitudes more uniformly Lin et al. (2024a). QuIP Chee et al. (2023) proposes an incoherence processing step based on orthogonal transformations, QuIP# Tseng et al. (2024) further improves this with randomized Hadamard transforms. QuaRot Ashkboos et al. (2024) applies Walsh-Hadamard Transformation (WHT) in Eq 2 to leverage the orthogonal invariance of models.

$$\boldsymbol{W}\boldsymbol{X} = \mathbf{H}^T \mathcal{Q}(\mathbf{H}\boldsymbol{W})\boldsymbol{X}, \quad \mathbf{H}_2 = \frac{1}{\sqrt{2}}\begin{bmatrix} 1 & 1 \\ 1 & -1 \end{bmatrix} \quad \text{and} \quad \mathbf{H}_{2^n} = \mathbf{H}_2 \otimes \mathbf{H}_{2^{n-1}}, \quad (2)$$

where $\mathcal{Q}$ denotes quantization. SpinQuant Liu et al. (2025) replaces the process with learnable rotation matrices.

(c) **Quantization Format:** Different quantization formats can be used according to the distribution, for instance, LQER adopts the `MaxInt` format Zhang et al. (2024a). Instead of scalar quantization, vector quantization uses codebooks to better approximate weight patterns. GPTVQ Van Baalen et al. (2024) improves GPTQ Frantar et al. (2023) by multidimensional vector quantization. AQLM Egiazarian et al. (2024) employs a learnable codebook, while QuIP# enhances codebook efficiency using the $E_8$ lattice structure Viazovska (2017).

**Solution to F2■: Preserving Accuracy for Important Activations.** The heavy-tailed nature of activation distributions suggests that precision for significant activations is crucial. This insight has led to the following strategies: A

(a) **Activation-Aware Weight Scaling:** AWQ Lin et al. (2024b) scales weights to preserve the quantization accuracy of weights corresponding to important activations and AffineQuant Ma et al. (2024) introduces affine transformation quantization for LLMs.

(b) **Low-Rank in quantization:** While scaling improves robustness for important weight channels, it may amplify quantization errors on less significant ones, especially in ultra-low-bit regimes. To address this, several methods apply low-rank decomposition quantization which can be formulated as:

$$i).\boldsymbol{W}\boldsymbol{X} = (\hat{\boldsymbol{W}} + (\boldsymbol{W} - \hat{\boldsymbol{W}})_r)\boldsymbol{X}; \quad ii).\boldsymbol{W}\boldsymbol{X} = (\boldsymbol{W}_r + \hat{\boldsymbol{R}})\boldsymbol{X}, \quad (3)$$

here $\hat{\boldsymbol{R}} = \mathcal{Q}(\boldsymbol{W} - \boldsymbol{W}_r)$ is the quantized residual matrix and $\boldsymbol{W}_r$ is rank $r$ matrix stored in high precision. There are two forms in the Eq 3, SVD-Quant Li et al. (2024) applies form $ii)$ and extends the approach to 4-bits diffusion models and the potential at low bits has not been explored, while fine-tuning PTQ methods apply form $i)$ such as LQER Zhang et al. (2024a) combines LoRC with MXINT Darvish Rouhani et al. (2020) quantization; RILQ Lee et al. (2025) and CALDERA Saha et al. (2024) further refine it by layer sensitivity analysis and low-bit iteration respectively.

We evaluate these methods across quantization accuracy ↑, costs↓, compression ratio↓, and inference latency↓. Those in ¶**O1.a**, ¶**F1.a**, and ¶**F2.a** exhibit lower quantization costs↓; however, they suffer noticeable accuracy degradation↓ below 3-bit quantization. In contrast, ¶**F1.bc** and ¶**F2.b** achieve stronger results↑ in low-bit but introduce additional inference latency↑. Low-rank methods in ¶**F2.b** even increase compression ratio↑. Furthermore, task-specific fine-tuning methods generally achieve higher accuracy↑ but lead to a longer runtime↑ and would compromise the model generalization↓. Therefore, Our objective is *"Designing a high-accuracy, fine-tuning-free quantization algorithm that simultaneously minimizes inference latency and additional memory overhead."*

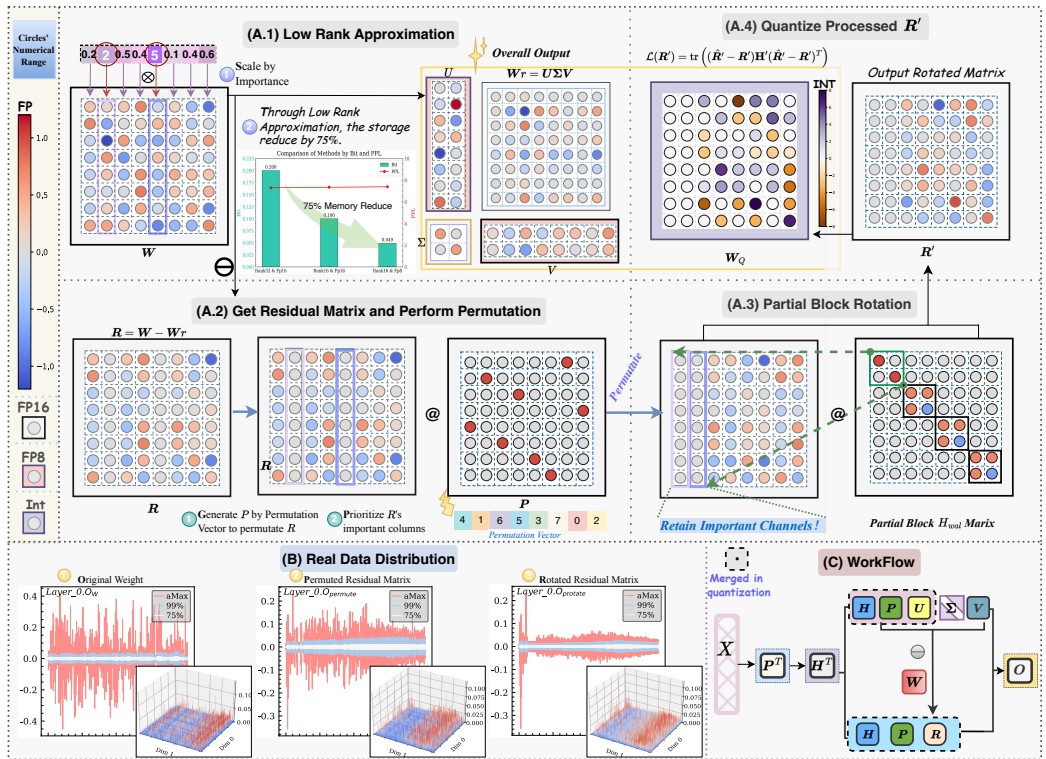

Figure 1: In LoPRo, the permutation-partial rotation effectively relieves the quantization burden of the residual matrix. Moreover, the R1SVD low-rank approximation maintains high efficiency and compression ratio. Specifics: (A). Using a real matrix and its variants to illustrate the data characteristics and transformations in LoPRo process;(B). A layer of corresponding empirical distribution in LoPRo and more visualization shown in Appendix K; (C). The Inference workflow of LoPRo.

## 3 METHOD

In this section, we first analyze the bottlenecks of low-rank methods in quantization accuracy. Then we propose Low-rank Partial Rotation Quantization – **LoPRo**, a novel fine-tuning-free PTQ method that leverages low-rank quantization and applies partial rotation to efficiently redistribute outliers in the residual matrix. Finally, we enhance quantization performance by incorporating an improved R1SVD algorithm. The workflow of LoPRo is presented in Figure 1.

### 3.1 PROBLEM STATEMENT

Previous methods suffer from suboptimal accuracy without fine-tuning due to insufficient utilization of the three key characteristics. The problem we aim to address is: *How to effectively balance three strategies to achieve the highest accuracy?*

We observe that the strategies in ¶**O1**, ¶**F1** are not mutually exclusive but rotation and truncation in ¶**F1** would disrupt the important channel preserved in ¶**F2.a**. Instead, the low-rank method is considered to be a promising direction. Specifically, the $W_r$ with high precision can be fused with rotation without introducing additional storage or computational overhead.

Between the two forms in Eq 3, the first is more suitable for fine-tuning in the LoRA scenario, as low-rank approximation is performed independently after quantization. However, for fine-tuning-free PTQs, we consider that the second form in Eq 3 is better, because the original $W$ is more amenable to low-rank approximation, as quantization disrupts its inherent low-rank structure. In this form, the low-rank matrix $W_r$ can be calculated by:

$$\{U', \Sigma, V\} = \text{SVD}(W\alpha), \ \ V' = V\alpha^{-1}, \ \ \ W_r = W_L W_R = (U\Sigma)(V'), \qquad (4)$$

where $\alpha$ is a scaling factor calculated by layer input. After applying low-rank decomposition with scaling, the property in ¶**F2** is utilized. Quantizing the residual matrix becomes a critical challenge. Since the low-rank approximation alone cannot fully smooth the value distribution within the weight matrix, a reasonable approach is to apply rotation to the residual matrix to further enhance accuracy. However, results in Table 5 show that applying a full Hadamard rotation degrades quantization accuracy (8.4 to 9.49 of perplexity↓ in LLaMA2-7B).

### 3.2 PARTIAL ROTATION QUANTIZATION UNDER PERMUTATION

For quantizing the residual matrix $\boldsymbol{R}$, two key observations are made: ***i).*** After token-wise scaling of the weights, more important columns exhibit smaller values, and maintaining their numerical stability is crucial for overall quantization accuracy. ***ii).*** It's difficult in quantization to remain columns determined by the diagonal of proxy Hessian.

We propose a block-wise partial rotation quantization algorithm for the residual matrix, which consists of the following three components:

**1. Column Permutation:** Considering quantization difficulty and column importance, we introduce a reordering strategy formulated as Eq 5, and leave a detailed discussion in Appendix A.4.

$$perm = sort\big(\text{diag}(\boldsymbol{H})/amean(\boldsymbol{R}, axis = 0)\big).idx \quad \text{and} \quad \boldsymbol{R} = \boldsymbol{W} - \boldsymbol{W}_r \;, \tag{5}$$

where $\boldsymbol{H}$ is the proxy Hessian computed in Eq 1, $\boldsymbol{R}$ is the residual matrix after low-rank approximation and $amean$ computes the mean of absolute values along the first dimension. This reordering can be expressed by applying permutation matrix $\boldsymbol{P}$, where $\boldsymbol{P}_{i,perm[i]} = 1$ and other elements are 0. The matrix $P$ is orthogonal and satisfied $\boldsymbol{P}^T\boldsymbol{P} = \boldsymbol{I}$. After permutation, important columns are moved to the leading columns of the residual matrix.

**2. Partial block Rotation:** Rotation enhances the incoherence between $\boldsymbol{R}$ and $\mathbf{H}$, thereby improving quantization accuracy. However, applying a full rotation would disrupt the column importance ordering and degrade accuracy. To address this, we propose a partial rotation in a block-wise manner:

$$\boldsymbol{Q} = \begin{bmatrix} \boldsymbol{I} & 0 & \cdots & \cdots & 0 \\ 0 & \mathbf{H}_{wal} & 0 & \cdots & \vdots \\ \vdots & 0 & \ddots & 0 & \vdots \\ \vdots & \vdots & 0 & \ddots & 0 \\ 0 & \cdots & \cdots & 0 & \mathbf{H}_{wal} \end{bmatrix}, \boldsymbol{R} = \boldsymbol{RPQ} \tag{6}$$

where $\boldsymbol{I}$ is an identity matrix of size $b_I$, and $\mathbf{H}_{wal} \in \mathbb{R}^{b_H \times b_H}$ is a Walsh-Hadamard matrix and $b_H$ is an integer power of 2. In this case, the leading columns, corresponding to the most important channel, are unrotated (identity block) to preserve them from degradation of quantization accuracy. The importance of other columns after permutation gradually decreases, as shown in Figure 1. By applying block-wise Walsh-Hadamard rotation of similarly important columns, it's easier to quantize the $\boldsymbol{R}'$ than $\boldsymbol{R}$. In this case, we have Theorem.1 and proved in Appendix B.2.

**Theorem 1 (Rotation).** *Denote $\mathcal{L}_{orig}$ as the original quantization loss, and $\mathcal{L}_{rot}$ as the loss under rotation in Eq 6; we deduce that*

$$\mathcal{L}_{\text{rot}}(\boldsymbol{R}) \leq \mathcal{L}_{\text{orig}}(\boldsymbol{R}) \tag{7}$$

**3. Quantization with Rotation Matrix**

After the low-rank approximation and partial Walsh-Hadamard rotation described in the previous section, we effectively leverage the feature ¶**F1** and ¶**F2** to obtain $\boldsymbol{R}'$ that is more amenable to quantization. Recall the low-rank quantization form in Eq 3 under transformation in Eq 6.

$$\boldsymbol{W}\boldsymbol{X} = \boldsymbol{W}_r\boldsymbol{X} + \mathcal{Q}(\boldsymbol{R}')\boldsymbol{Q}^T\boldsymbol{P}^T\boldsymbol{X}, \tag{8}$$

This formulation shows that the quantized component and the low-rank component are decoupled. Therefore, we can apply methods from ¶**O1** independently to improve the quantization of the $\boldsymbol{R}'$. The optimization function can be expressed as Eq 9 and the proof is given in Appendix B.3:

$$\mathcal{L}(\boldsymbol{R}) = \|\boldsymbol{R}\boldsymbol{X} - \hat{\boldsymbol{R}}\boldsymbol{X}\|^2 = \text{tr}\left((\hat{\boldsymbol{R}} - \boldsymbol{R})\boldsymbol{H}'(\hat{\boldsymbol{R}} - \boldsymbol{R})^T\right), \tag{9}$$

where $H' = Q^T P^T H P Q$ and $H$ is the origin proxy Hessian. In addition to this, quantization methods can be freely replaced with other advanced schemes, such as vector-wise quantization.

### 3.3 A Light Low-Rank Algorithm by Randomized Sketching

A common practice of low-rank methods preserves $W_L$ and $W_R$ in full precision Lee et al. (2025); Zhang et al. (2024a), while in CALDERA Saha et al. (2024), the low-rank components are further compressed by quantization to reduce memory costs. But this method requires a higher rank like 256 which incurs a considerable computational burden.

Therefore, to minimize the additional overhead introduced by low-rank decomposition while maintaining inference performance, we proposed **R1SVD**: a simplification to the randomized SVD algorithm Frieze et al. (2004); Musco & Musco (2015) under the rank-1 condition. This simplification introduces a rank-1 matrix approximation technique, as described below:

Given a matrix $A \in \mathbb{R}^{m \times n}$. For a standard RSVD (Randomized Singular Value Decomposition) prototype, it typically consists of the following two steps:
**Stage A:** Generate an $\mathbb{R}^{n \times r}$ Gaussian test matrix $S$ and form $Y = (AA^*)^{it} AS$, where $it$ is the iteration times. Construct a matrix $Q = QR(Y)$ by QR decomposition whose columns form an orthonormal basis of $Y$.

**Stage B:** Form $B = Q^* A$. Compute an SVD of the small matrix: $B = U' \Sigma V^*$. Set $U = QU'$.

If a rank-1 matrix $S \in \mathbb{R}^{n \times 1}$ is utilized for low-rank approximation of a matrix and substituted into the two stages, we also have $Y = (AA^*)^{it} AS$, then for the matrix $Y \in \mathbb{R}^{m \times 1}$, the QR decomposition can be directly represented as follows.

$$Q = \frac{Y}{\|Y\|} \in \mathbb{R}^{m \times 1}, R = \|Y\| \in \mathbb{R}^{1 \times 1}. \tag{10}$$

Similarly, the SVD decomposition for rank-1 matrix $B = Q^* A$ can be represented as follows:

$$U' = \{1\}, \Sigma = \|B\|, V = \frac{B}{\|B\|}. \tag{11}$$

Applying **Stage B** and Equation.10, we have the rank-1 matrix $A_1 = U\Sigma V$:

$$U = \frac{(AA^*)^{it} AS}{\|(AA^*)^{it} AS\|}, \quad \Sigma = \frac{\|S^* A^* (AA^*)^{it} A\|}{\|(AA^*)^{it} AS\|}, \quad V = \frac{S^* A^* (AA^*)^{it} A}{\|S^* A^* (AA^*)^{it} A\|}. \tag{12}$$

Then, we set $A = A - A_1$ and apply iteration to this process, which enables decomposition at any rank. Since the singular value matrix $\Sigma$ is diagonal with low storage cost, by storing $U$ and $V$ in *fp8* while retaining $\Sigma$ in *fp16*, the storage overhead can be reduced by half. Furthermore, the loss in precision is compensated in subsequent iterations, thereby preserving the overall accuracy.

### 3.4 Algorithm Analysis

In this section, we analyze LoPRo from:

- **Compression Ratio:** Consider a weight matrix $W \in \mathbb{R}^{m \times n}$ with an original precision $d_o$ (in bits), quantized to a target precision $d_q$ with group size $g$. The low-rank components of rank $r$ are stored in precision $d_r$. Then, the overall average compression bit $d_C$ is given by:

$$d_C = \overbrace{d_q + \frac{d_o}{g}}^{Quant\ W} + \overbrace{\frac{rd_r}{n} + \frac{rd_r}{m}}^{U\ and\ V} + \overbrace{\frac{2d_o}{n}}^{\alpha\ and\ P} + \overbrace{\frac{rd_o}{mn}}^{\Sigma}. \tag{13}$$

  We provide a detailed analysis and proof in Appendix A.2.

- **Inference Latency:** Assume the weight is square and the input is $\mathbf{X} \in \mathbb{R}^{n \times b}$. Therefore, the total inference latency complexity $C$ is shown in Eq 13 and detailed in Appendix A.3:

$$C = \mathcal{O}\big(nb(2r + 1 + \log b_H)\big). \tag{14}$$

  This shows that the additional latency grows linearly with the batch size $b$ and is manageable for small $r$ and moderate $b_H$, making the method suitable for efficient deployment.

## 4 EVALUATION

### 4.1 EXPERIMENT SETUP

**Models:** Evaluations are carried out on the LLaMA-2 and LLaMA-3 dense model families Touvron et al. (2023) and Mixture of Experts (MoE) model Mixtral-8x7B Jiang et al. (2024).

**Baseline:** Following the classification introduced in §2, we conduct a comprehensive comparison with the state-of-the-art (SOTA) PTQs within each class. Fine-tuning-free methods include: GPTQ Frantar et al. (2023), GPTVQ Van Baalen et al. (2024), OmniQuant Shao et al. (2023), QuIP# Tseng et al. (2024), LQER Zhang et al. (2024a), and MoEQuant Chen et al. (2025) [1]. Additionally, we compare with fine-tuning-based methods such as QuIP# and CALDERA Saha et al. (2024). These methods are categorized according to §2, with tags ▮ stand for ¶**O1.a,b**; ▮ stand for ¶**F1.a,b,c**; and ▮ stand for ¶**F2.a,b**, while ▮ indicates the unused of such a strategy [2].

**Metrics:** We conducted perplexity experiments on the WikiText2 Merity et al. (2016) and zero-shot experiments using test sets including ARC-challenge (AC), ARC-easy (AE)Boratko et al. (2018), PIQA (QA)Bisk et al. (2020), and Winogrande (WI)Sakaguchi et al. (2021), with the lm-evaluation-harness Gao et al. (2024) framework employed for testing. Other settings are detailed in Appendix.C.

### 4.2 MAIN RESULTS

We conduct a comprehensive comparison of LoPRo with several SOTA baselines on the LLaMA model family. The experimental results demonstrate that our proposed LoPRo consistently and notably outperforms all other baselines across various settings. Further analysis of these results can be categorized as follows:

**2-bit Scenario:** ❶. Compare with scalar quantization: LoPRo achieves better performance than GPTQ and as well as outperforming clipping-based approaches such as OmniQuant. Compared to LQER, which requires a significantly higher rank (e.g., $r = 256$) in the 2-bit regime, resulting in substantial memory overhead, LoPRo achieves better accuracy with only $r = 16$. In contrast, our method surpasses LQER in both quantization accuracy and average compression ratio. This improvement comes from the apply an importance-aware rotation strategy to capture the characteristic of residual components, thereby enhancing the overall reconstruction fidelity and quantization performance. ❷. Compare with vector quantization: We observe that vector quantization techniques generally outperform scalar methods in the 2-bit regime. Compared with GPTVQ, LoPRo achieves a perplexity reduction of **0.36↓** on the 7B model, along with approximately **4%** improvement in zero-shot accuracy, while introducing only 3% additional memory overhead. Furthermore, against QuIP#, LoPRo shows over **20%** Δ PPL improvement across all three tested models, with less than **10%** increase in memory usage.

**3-bit Scenario:** In the 3-bit setting, LoPRo consistently outperforms all fine-tuning-free baselines. Interestingly, we observe that vector quantization underperforms scalar quantization at 3-bit, and scalar-LoPRo achieves the best overall accuracy. We attribute this to the higher error tolerance at 3-bit, where simpler scalar quantization suffices. Moreover, in GPTVQ, 4d codebook usage at 3-bit leads to excessive memory consumption, forcing the use of 2d codebooks setups, which limits representational capacity and results in inferior reconstruction. In contrast, our method leverages structural decomposition and targeted rotation to maintain high accuracy without relying on complex codebooks.

**LLaMA-3 & MoE Results:** LLaMA-3 features with less redundant parameterization and more sensitive to low-bit quantization, leading to significant performance degradation in low-bit quantization Huang et al. (2024), methods such as clipping introduce more disruptive perturbations to the weight distribution, exacerbating accuracy loss. The results from Table 1 show that LoPRo achieves even better relative performance in LLaMA-3 compared to LLaMA-2. In the MoE model Mixtral-8x7B, LoPRo achieves equally significant results. Compared to GPTQ, LoPRo obtains up to a **10%** improvement in zero-shot accuracy. Notably, when evaluated on the MoE model Mixtral-

---

[1]The implementation of MoEQuant is now unavailable, it does not report results on several datasets used in our main experiments. To ensure a comprehensive comparison, we provide additional evaluation in Appendix E.

[2]For example, GPTVQ with ▮▮ ▮ means it employ loss method in ¶**O1.a** and vector quantization in ¶**F1.c**.

Table 1: The Quantization Results of LoPRo and baselines. We report perplexity of WikiText2 and four zero-shot accuracy. MoEQ is in short of MoEQuant++, see 1. The meaning of 'Tag' and abbreviations for zero-shot tasks follow §4.1. More detailed settings are given in Appendix C.

| Model | Method | Tag | Bit | PPL↓ | ZeroShot Acc↑ | | | | Bit | PPL↓ | ZeroShot Acc↑ | | | |
|---|---|---|---|---|---|---|---|---|---|---|---|---|---|---|
| | | | | | AC | AE | WI | QA | | | AC | AE | WI | QA |
| LLaMA2-7B | FP16 | | 16 | 5.12 | 43.4 | 76.3 | 69.1 | 78.4 | 16 | 5.12 | 43.4 | 76.3 | 69.1 | 78.4 |
| | GPTQ | | 2.13 | 50.8 | 20.9 | 34.9 | 52.3 | 57.2 | 3.13 | 8.06 | 31.1 | 58.5 | 59.2 | 71.5 |
| | GPTVQ | | 2.13 | 6.89 | 30.2 | 64.3 | 64.1 | 72.1 | 3.13 | 5.61 | 39.9 | 74.1 | **69.1** | 76.2 |
| | LQER | | 2.80 | 10.3 | 33.1 | 60.4 | 61.2 | 70.7 | 3.28 | 5.72 | 40.4 | 74.2 | 68.4 | 74.9 |
| | OmniQ | | 2.13 | 15.0 | 28.8 | 58.1 | 59.1 | 70.2 | 3.13 | 5.81 | 40.8 | 74.5 | 67.5 | 77.7 |
| | Quip# | | 2.00 | 8.22 | 29.9 | 61.3 | 61.7 | 69.6 | 3.00 | 5.79 | 40.2 | **75.1** | 67.0 | 76.2 |
| | LoPRo | | 2.17 | 7.39 | 31.2 | 62.8 | 63.8 | 71.1 | 3.17 | **5.43** | **41.0** | 74.9 | 68.9 | 76.3 |
| | LoPRo$_v$ | | 2.17 | **6.53** | **34.6** | **69.0** | **66.5** | **72.7** | 3.17 | 5.45 | **41.0** | 74.8 | **69.1** | 76.7 |
| LLaMA2-13B | FP16 | | 16 | 4.57 | 49.1 | 77.4 | 73.9 | 81.4 | 16 | 4.57 | 49.1 | 77.4 | 73.9 | 81.4 |
| | GPTQ | | 2.13 | 43.8 | 23.3 | 43.3 | 54.7 | 61.3 | 3.13 | 5.85 | 38.5 | 65.7 | 63.9 | 76.5 |
| | GPTVQ | | 2.13 | 5.78 | 38.7 | 73.6 | **68.5** | **75.4** | 3.13 | 4.92 | 44.5 | 75.2 | **72.0** | 77.8 |
| | LQER | | 2.64 | 8.42 | 33.2 | 65.8 | 66.4 | 73.1 | 3.24 | 5.12 | 42.3 | 76.7 | 71.2 | 77.4 |
| | OmniQ | | 2.13 | 11.1 | 31.3 | 62.3 | 65.6 | 72.3 | 3.13 | 5.11 | 42.0 | 77.9 | 71.3 | 78.0 |
| | Quip# | | 2.00 | 6.06 | 36.2 | 68.6 | 63.6 | 74.2 | 3.00 | 4.90 | 42.2 | 76.6 | 71.5 | 77.6 |
| | LoPRo | | 2.17 | 6.48 | 33.6 | 69.0 | 66.3 | 72.4 | 3.17 | **4.84** | 44.9 | **78.7** | 71.1 | **78.5** |
| | LoPRo$_v$ | | 2.17 | **5.79** | **38.8** | **74.2** | 68.0 | **75.4** | 3.17 | 4.87 | 44.5 | 77.4 | 71.0 | 78.3 |
| LLaMA3-8B | FP16 | | 16 | 5.54 | 50.2 | 80.1 | 73.5 | 79.7 | 16 | 5.54 | 50.2 | 80.1 | 73.5 | 79.7 |
| | GPTQ | | 2.13 | 2e2 | 21.1 | 29.3 | 52.1 | 54.4 | 3.13 | 7.81 | 37.7 | 70.5 | 71.1 | 74.9 |
| | GPTVQ | | 2.13 | 9.32 | 31.3 | 57.3 | **68.3** | 67.8 | 3.13 | 6.78 | 45.1 | 76.7 | 72.5 | 77.8 |
| | OmniQ | | 2.13 | 54.1 | 19.3 | 36.1 | 51.9 | 59.0 | 3.13 | 7.01 | 42.2 | 72.4 | 71.3 | 75.5 |
| | Quip# | | 2.00 | 10.9 | 30.8 | 57.1 | 67.0 | 67.5 | 3.00 | 6.75 | 45.2 | 75.2 | 72.3 | 78.0 |
| | LoPRo | | 2.17 | 10.8 | 30.5 | 58.9 | 62.2 | 68.2 | 3.17 | **6.31** | 44.1 | 75.9 | **73.0** | 77.5 |
| | LoPRo$_v$ | | 2.17 | **8.95** | **37.0** | **69.2** | 68.0 | **71.4** | 3.17 | 6.41 | **45.3** | **76.8** | 71.9 | **78.4** |
| Mixtral-8x7B | FP16 | | 16 | 3.84 | 62.0 | 87.3 | 75.3 | 83.5 | 16 | 3.84 | 62.0 | 87.3 | 75.3 | 83.5 |
| | GPTQ | | 2.13 | 14.1 | 26.6 | 35.7 | 49.5 | 57.7 | 3.13 | 4.71 | 52.1 | 69.3 | 74.4 | 80.9 |
| | GPTVQ | | 2.13 | 5.28 | 42.0 | 71.6 | 66.5 | 75.9 | 3.13 | 4.27 | 54.9 | 72.9 | 74.8 | 82.6 |
| | MoEQ | | 3.00 | 4.90 | - | - | - | - | 3.00 | 4.90 | - | - | - | - |
| | LoPRo | | 2.17 | 5.25 | 53.2 | 81.8 | 72.0 | 78.1 | 3.17 | **4.15** | **61.0** | 85.3 | **77.0** | **83.3** |
| | LoPRo$_v$ | | 2.17 | **4.80** | **55.8** | **82.9** | **74.0** | **80.5** | 3.17 | 4.16 | 60.6 | **86.5** | 76.7 | 82.9 |

8x7B, LoPRo achieves better performance at an average bitwidth of 2.17 than MoEQuant++ at 3-bit precision! The results demonstrate the strong effectiveness and generalization of our algorithm.

**Fine-tuning Results:** As LoPRo presents a general PTQ method that achieves strong performance within once quantization, it can be further enhanced through integration with fine-tuning methods. We compare RILQ with LoPRo against RILQ and CALDERA with QuIP#. Results in Table 2 demonstrate that LoPRo achieves better performance than fine-tuning Quip# across all bitwidths, which highlighting the extensibility of LoPRo. Moreover, the **3-bit fine-tuning-free** results in Table 1 are very close to here (only **0.06** PPL degradation on the LLaMA-2 7B and **0.04** on 13B model). This indicates that, at 3-bit quantization, the partial-block-wise rotation strategy can eliminate the majority of quantization error, making it fine-tuning-free to achieve strong performance.

## 4.3 QUANTIZATION COST

We report the runtime comparison in Table 3. In LoPRo, the quantization costs primarily consist of ❶ low-rank sketching and block-wise $H_{wal}$ transformation to $R$; ❷ the quantization of $\mathbf{R}'$. Compared to methods such as OmniQuant and QuIP#, LoPRo is significantly faster that requiring less than **2.5** hours to quantize the Mixtral-8x7B model. This efficiency stems from the employ of R1SVD for low-rank decomposition, which operates in $\mathcal{O}(N^2)$ time complexity, growing linearly with model size, in contrast to the $\mathcal{O}(N^3)$ cost of full SVD; Notably, even under vector quantization,

Table 2: Fine-tuning results for the LLaMA-2 family. The notation "rank(bit)" denotes the rank and bitwidth used for the low-rank components. LoPRo employs RILQ as the fine-tuning backend; further implementation details are provided in Appendix C.

| | Method | Tag | Bit | rank(bit) | PPL | ZeroShot Acc | | | |
|---|---|---|---|---|---|---|---|---|---|
| | | | | | | AC | AE | WI | QA |
| LLaMA2-7B | CALDERA | | 2.2 | 128(4) | 6.79 | 34.6 | 65.1 | 63.8 | 75.1 |
| | RILQ | | 2.2 | 32(16) | 6.28 | 37.4 | 70.1 | **66.5** | 75.0 |
| | LoPRo$_v$-FT | | 2.2 | 32(8) | **6.06** | 38.1 | 70.9 | 64.8 | 76.1 |
| | RILQ | | 3.2 | 32(16) | 5.47 | 40.6 | 75.8 | 67.9 | 76.9 |
| | LoPRo$_v$-FT | | 3.2 | 32(8) | **5.37** | 42.8 | 76.0 | 68.9 | 77.2 |

| | Method | Tag | Bit | rank(bit) | PPL | ZeroShot Acc | | | |
|---|---|---|---|---|---|---|---|---|---|
| | | | | | | AC | AE | WI | QA |
| LLaMA2-13B | CALDERA | | 2.2 | 128(4) | 5.72 | 38.7 | 68.5 | 67.9 | 76.0 |
| | RILQ | | 2.2 | 32(16) | 5.42 | 40.1 | 71.2 | **68.8** | 78.1 |
| | LoPRo$_v$-FT | | 2.2 | 32(8) | **5.34** | 42.8 | 75.6 | 68.5 | 78.9 |
| | RILQ | | 3.2 | 32(16) | 4.88 | 45.4 | 76.4 | 71.4 | 79.2 |
| | LoPRo$_v$-FT | | 3.2 | 2(8) | **4.80** | 46.2 | 77.9 | 71.7 | 80.1 |

Table 3: Quantization time for methods, where 'h' denotes hours and 'm' denotes minutes. *NA* indicates that the method was not implemented or encountered OOM errors during execution.

| Model | GPTQ | GPTVQ | LQER | AQLM | OminiQ | LoPRo | LoPRo$_v$ |
|---|---|---|---|---|---|---|---|
| LLaMA2-7B | 25.2m | 1.5h | 45.2m | 11.1h | 3.1h | **26.4m**↓↓ | **32.2m**↓ |
| LLaMA2-13B | 40.5m | 3.7h | 1.2h | 22.7h | 5.3h | **44.5m**↓↓ | **56m**↓ |
| Mixtral-8x7B | 2.6h | *NA* | *NA* | *NA* | *NA* | **2.0h**↓↓ | **2.4h**↓ |

LoPRo remains faster than GPTVQ. This is because we adopt only the simplest form of vector quantization—without advanced components such as LoRA or fine-tuning and even achieve higher accuracy. This further demonstrates the effectiveness and practicality of our approach.

## 4.4 INFERENCE EFFICIENCY

In inference, we take the W4A16 kernel in GPTQModel qubitium (2024) as a baseline and the throughput and latency of LoPRo are shown in Table 4. Since only linear layers are affected, the rotation can be efficiently applied to the input $\mathbf{X}$ by Fast Walsh-Hadamard Transform in $\mathcal{O}(nlog(n))$ Tseng et al. (2024). Furthermore, the low-rank components are computed with a very small rank ($r = 16$) and

Table 4: Inference throughput and latency of the LoPRo.

| Model | Batch | Decode (token/s) | | Latency | | |
|---|---|---|---|---|---|---|
| | | Baseline | LoPRo | Rotation | Low-rank | Total |
| 7b | 1 | 93.3 | 83.7 | 4.3% | 6.0% | 10.3% |
| | 16 | 368.6 | 334.5 | 3.7% | 5.6% | 9.3% |
| | 64 | 450.0 | 412.8 | 3.2% | 5.1% | 8.3% |
| 13b | 1 | 62.4 | 56.1 | 3.9% | 6.2% | 10.1% |
| | 16 | 274.4 | 252.4 | 3.3% | 4.7% | 8.0% |
| | 64 | 388.4 | 358.2 | 2.9% | 4.9% | 7.8% |

only brings below 10% latency, and this overhead further decreases as model scale and batch size increase—aligning well with the theoretical analysis presented in Section 3.4.

## 4.5 ABLATION STUDIES

We present the performance in different components at the 2-bit level in Table 5. Models using simple RTN (Round-To-Nearest) within scaled low-rank quantization exhibit severe performance degradation, and can be significantly improved by minimizing proxy loss. However, when full $\boldsymbol{H}_{wal}$ transformation is further applied, the accuracy drops to 51.5%, indicating that excessive rotation disrupts the importance structure established by scaling LoRA. In contrast, changing it to partial block $\boldsymbol{H}_{wal}$ with permutation can reduce perplexity by 10% and increasing accuracy by 5%. This validates the critical role of synergistic optimization between residual quantization and structured rotation, consistent with the observations and analyzes in § 3. Furthermore, upgrading the quantization scheme to vector quantization (VQ) reduces PPL further to 6.53, achieving the best performance among

Table 5: Ablation on different components in LoPRo under 2-bit LLaMA2-7b. 'NA' means non-use of such strategy. 'OQ' and 'VQ' denote use GPTQ and GPTVQ as the quantizer respectively. The last two bolded lines represent LoPRo and LoPRo$_v$

| Bits | Rotation | Quant | Tag | PPL | Avg.acc |
|------|----------|-------|-----|-----|---------|
| 16 | NA | NA | | 5.11 | 66.8 |
| 2.2 | NA | RTN | | 4.0e2 | 44.1 |
| 2.2 | NA | OQ | | 8.4 | 53.0 |
| 2.2 | Full $H_{wal}$ | OQ | | 9.49 | 51.5 |
| 2.2 | Partial $H_{wal}$ | OQ | | 7.39 | 57.8 |
| 2.2 | Partial $H_{wal}$ | VQ | | 6.53 | 61.2 |

2-bit quantization methods. This highlights the high extensibility and composability of the proposed low-rank rotation framework in LoPRo. In addition, ablation on other parameters are given in Appendix D.

## 5 ADDITIONAL EXPERIMENTS

To ensure a fair comparison with the MoE baseline and to validate the performance of LoPRo on updated model architectures, we conducted comprehensive experiments following the baseline setup in MoEQuant; the results are presented in Appendix E. Additionally, we evaluated our method on two model architectures—Qwen2.5 and Qwen3—with detailed results reported in Appendix F. Furthermore, for the Qwen3 model, we performed extensive evaluations on the Open LLM Leaderboard V1 to assess the degradation introduced by quantization; these results are provided in Appendix G.

The additional results consistently align with those from our main experiments: LoPRo achieves strong performance under high-compression quantization across diverse model architectures and scales. Notably, under 3-bit quantization, LoPRo's scalar-level algorithm demonstrates near-lossless behavior on the OpenLLM benchmark. Collectively, these experiments highlight the excellent scalability of LoPRo.

## 6 CONCLUSION

In this work, we propose LoPRo, an efficient PTQ method that utilizes the characteristics of the residual matrix after low-rank decomposition. LoPRo introduces a block-wise partial Hadamard rotation under permutation, which effectively reduces the quantization difficulty of the residual matrix while preserving its importance structure. Furthermore, we employ a mixed-precision R1SVD approximation to replace SVD, significantly reducing computational error and accelerating decomposition. Through comprehensive evaluations, LoPRo achieves state-of-the-art performance across various tasks including MoE LLMs, achieving high accuracy, compression ratio and efficiency in fine-tuning-free quantization while preserving scalability for fine-tuning.

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

CONTENTS

Table 6: Symbols and Description in this paper.

| Symbols | Description |
|---|---|
| $\boldsymbol{W}$ | The weight matrix $\boldsymbol{W} \in \mathbb{R}^{m \times n}$. |
| $\boldsymbol{X}$ | The input of each linear layer |
| $\mathcal{Q}$ | Quantization function |
| $\hat{\boldsymbol{W}}$ | A pseudo quantized matrix |
| $\boldsymbol{H}$ | A proxy hessian equal to $E_{\boldsymbol{X}}[\boldsymbol{X}\boldsymbol{X}^T]$ |
| $\mathbf{H}$ | A Hadamard matrix |
| $\boldsymbol{W}_r = \boldsymbol{U}\boldsymbol{\Sigma}\boldsymbol{V}$ | The rank-r approximate matrix of $\boldsymbol{W}$ |
| $\mathcal{L}(\cdot)$ | The loss function, we use the proxy Hessian in this paper. |
| $\boldsymbol{R}$ | The residual matrix of low-rank approximation $\boldsymbol{R} = \boldsymbol{W} - \boldsymbol{W_r}$ |
| $\mathbf{H}_{wal}$ | Walsh-Hadamard matrix |
| $\boldsymbol{Q}$ | The partial block Walsh-Hadamard matrix |
| $\boldsymbol{P}$ | The permutation matrix from index vector $perm$. |
| $\boldsymbol{I}$ | An identity matrix |
| $\boldsymbol{E}$ | The error matrix under quantization $\boldsymbol{E} = \boldsymbol{W} - \hat{\boldsymbol{W}}$. |
| $\mathcal{O}(\cdot)$ | The upper bound of time complexity |
| $sort(\cdot).idx$ | Sort array in ascending order and take the index. |
| $amean(\cdot, axis = 0)$ | Take the average of the absolute values along the dimension 0. |
| $\boldsymbol{E}_j$ | $j$-th column of the error matrix $\boldsymbol{E}$. |
| $\langle \boldsymbol{E}_j, \boldsymbol{E}_l \rangle_{\boldsymbol{\tau}}$ | Weighted inner product between error vectors $\boldsymbol{E}_j$ and $\boldsymbol{E}_l$. |
| $\mathrm{Cov}_{\boldsymbol{\tau}}(j, l)$ | Weighted covariance between columns $j$ and $l$ of $\mathbf{W}$. |
| $\mathrm{Var}_{\boldsymbol{\tau}}(j)$ | Weighted variance of the $j$-th column |

## A  LoPRo Details:

In this section, we provide a detailed description of the LoPRo execution pipeline,

### A.1  Algorithm pseudo-code

The workflow of LoPRo is outlined in Algorithm 1:

**Stage A. Low-Rank Approximation under Calibration Set:**

1. Perform forward inference on the calibration dataset to collect layer-wise input activations $\boldsymbol{X}$ corresponding to each weight matrix $\boldsymbol{W}$ (line 1-4).

2. Compute the importance-aware scaling vector with the activations $\boldsymbol{X}$ (line 5-6).

3. Apply the scaling to $\boldsymbol{W}$, obtaining scaled weight (line 7).

4. Draw a sketch vector and form exponentiation to the Gram matrix (line 8-10).

5. Calculate the rank-1 matrix according to Eq 12, and update the matrix for low-rank approximation (line 11-14).

**Stage B. Structured Residual Rotation:**

6. Compute the permutation array according to Eq 5, which groups columns by importance; then construct the corresponding permutation matrix $\boldsymbol{P}$ (line 15-17).

---

**Algorithm 1:** Flexible Low-Rank Matrix Sketching Quantization

---

**Data:** $Module$(module weight), $Sample$(calibration data), $d$(quantization bit),$rank$(the rank)

**Result:** $QModule$(quantized module weight), $LoraModule$(low-rank component)

1 **for** $layers$ $in$ $Module$ **do**

2     Obtain the activation for each layer during model inference: $Acts \leftarrow layers.forward().$;

3     **for** $l$ $in$ $layers$ **do**

4        Obtain the weights and corresponding activation in Acts:
         $\{\boldsymbol{W}, \boldsymbol{X}\} \leftarrow \{layers[l], Acts[l]\}$;

5        Calculate the mean of the activation values: $\overline{X} = mean(|\boldsymbol{X}|, axis = 0)$;

6        Calculate the scale from $\boldsymbol{X}$: $s \leftarrow \overline{X}^{2.5}/\sqrt{max(\overline{X}) * min(\overline{X})}$;

7        Get $\boldsymbol{W}_s$: $\boldsymbol{W}_s \leftarrow \boldsymbol{W} \cdot diag(s)$;

8        **for** $r$ $in$ $rank$ **do**

9           draw a random sketch vector: $v = random(\boldsymbol{W}.shape[1])$;

10          calculate $2 + iter$ times $GEMV$: $y = (\boldsymbol{W}_s\boldsymbol{W}_s^T)^{iter}\boldsymbol{W}_s v, \ p = \boldsymbol{W}_s^T y$;

11          Get rank-1 component:
            $U_1 = (y/\|y\|).to(fp8) \ , \ V_1 = (p/\|p\|).to(fp8) \ , \ \sigma = \|p\|/\|y\|$;

12          Add to the low-rank component: $\boldsymbol{U}.add(U_1), \boldsymbol{V}.add(V_1), \boldsymbol{\Sigma}.add(\sigma)$;

13          Update the matrix: $\boldsymbol{W}_s = \boldsymbol{W}_s - \sigma U_1 V_1$;

14        **end**

15        Calculate Residual matrix and proxy Hessian: $\boldsymbol{R} = diag(s)^{-1} \cdot \boldsymbol{W}_s \ , H = E_X[XX^T]$ ;

16        Get permutation index p: $p = sort(diag(\boldsymbol{H})/amean(\boldsymbol{R}, dim = 0)).idx$;

17        Build permutation matrix: $\boldsymbol{P} = 0, \boldsymbol{P}_{k,p[k]} = 1$;

18        Apply partial block hadamard transformation in Eq 6: $\boldsymbol{R}' = \boldsymbol{R}\boldsymbol{P}\boldsymbol{Q}$;

19        Quantize the resident matrix by methods like GPTQ/GPTVQ... by new loss:
         $\boldsymbol{W}_q = \mathcal{Q}(\boldsymbol{R}') \ , \mathcal{L}(\boldsymbol{R}) = \text{tr}\left((\hat{\boldsymbol{R}} - \boldsymbol{R})\boldsymbol{H}'(\hat{\boldsymbol{R}} - \boldsymbol{R})^T\right), \ \boldsymbol{H}' = \boldsymbol{Q}^T\boldsymbol{P}^T\boldsymbol{H}\boldsymbol{P}\boldsymbol{Q}$;

20        Add quantization results:$qlayer.add(\boldsymbol{W}_q, \boldsymbol{U}, \boldsymbol{V}, \boldsymbol{\Sigma}, p, s)$;

21     **end**

22     Save the layer result: $QModule.layer[name] = qlayer$

23 **end**

24 **return** $QModule$;

---

7. Apply the block-wise partial Hadamard transformation to the residual matrix to minimize quantization error (line 18).

**Stage C. Quantization of Transformed Residual:**

8. Quantize the rotated residual matrix $\boldsymbol{R}'$ by quantization tools (e.g., GPTQ for scalar quantization or GPTVQ for vector quantization) (line 19).

9. Save the final quantized components: $\boldsymbol{W}_q$, $\boldsymbol{U}$, $\boldsymbol{\Sigma}$, $\boldsymbol{V}$, scaling vector $\boldsymbol{a}$, and permutation index $\boldsymbol{p}$, for deployment (line 20-22).

A.2   AVERAGE BIT-WIDTH

For a weight matrix $\boldsymbol{W} \in \boldsymbol{R}^{m \times n}$ with original precision $d_o$ (in bits) and quantized to target $d_q$ with group size $g$. Suppose the low-rank components of rank $r$, stored in precision $d_r$.

- For the quantized part, the bit-width of matrix $W_q$ is $d_q$, and the average bit of scale is $\frac{d_o * m * n/g}{m*n}$. Then the average bits in this part is $d_q + \frac{d_o}{g}$.

- For the low-rank matrix: $\boldsymbol{U}$ and $\boldsymbol{V}$ are stored in $d_r$ costs $\frac{d_r * (m+n)}{m*n}$. The singular is a diagonal matrix and can transform to a vector of size $r$ that costs $\frac{r*d_0}{m*n}$.

- For the permutation $p$ and scale vector $s$ in 1, the average bit is both $\frac{m*d_0}{m*n}$.

Combining these three items, we have the form in Eq 13. Specifically, the last two terms in Eq 13 are negligible while $r$ and $d_o$ are far less than weight dimension, . In practice, we set the rank $r$ to 16 or 32 and store $U$ and $V$ in *fp8* precision. For a 7B model, it introduces an additional storage overhead of approximately 0.05-bits and 0.04-bits in a 13B model. This overhead further diminishes as model scale increases, demonstrating the algorithm's high compression efficiency and scalability.

### A.3 TIME COMPLEXITY

The algorithm introduces two main part of latency during inference: low-rank matrix multiplication and reordering with rotation transformations. Assume the linear weight dimension is equal, and the input is $X \in \mathbb{R}^{n \times b}$.

- For the original quantization linear layer $O = Dequant(W)X$. The complexity is $\mathcal{O}((b + 1)n^2)$.

- For the low-rank part, we first $Y = VX$ followed by $O = U(\Sigma Y)$. Since $\Sigma$ is diagonal, its multiplication takes linear time, and the overall complexity is $\mathcal{O}(2rnb)$.

- For the permutation and rotation part, we apply it to the input $X$ to minimize computational overhead: $\tilde{X} = Q^T P^T X$. The permutation has complexity $\mathcal{O}(nb)$ and the rotation can be computed efficiently by the Fast Walsh-Hadamard Transform Fino & Algazi (1976) since the block size $b_H = 2^i$, running in $\mathcal{O}(nb \log b_H)$.

Together, we have the total complexity $C$ in Eq 14.

### A.4 PERMUTATION

The construction of the rearrangement formula in Eq 5 is motivated by the following two key observations:

- For the residual matrix $R$, after scaled low-rank approximation, columns corresponding to more important weight channels are approximated with higher accuracy, resulting in smaller absolute values. In other words, columns in $R$ with smaller absolute values correspond to more critical channels; preserving the quantization precision of these top-ranked important columns is crucial for overall accuracy. Sorting columns by $1/amean(R, axix = 0)$ (where amean denotes average magnitude) places relatively more important columns at the front, allowing subsequent application of the identity matrix to avoid uniform quantization degradation across all columns.

- In loss optimization methods such as GPTQ, it is preferable to first quantize columns with larger quantization errors, followed by those with smaller errors. This ordering is determined by the diagonal elements of the Hessian matrix, diag($H$).

Combining these two principles, we derive the final column ranking function as presented in Eq 5. Under the permutation, we can preserve the quantization precision of the most critical columns while jointly optimizing the quantization of smoothly varying and less critical columns, thereby achieving superior overall quantization performance.

## B PROOFS

### B.1 NOTATIONS AND ASSUMPTIONS

**Notation 1. Original weight**: Let the weight matrix:

$$W = [\mathbf{w}_1, \mathbf{w}_2, \ldots, \mathbf{w}_n] \in \mathbb{R}^{m \times n}. \tag{15}$$

has mean $\mu$ and variance $\sigma$ and each column $\mathbf{w}_j \in \mathbb{R}^m$. In $j$-th column:

$$\mu_j = \frac{1}{m} \sum_{i=1}^{m} w_{ij}, \quad \sigma_j^2 = \frac{1}{m} \sum_{i=1}^{m} (w_{ij} - \mu_j)^2 \tag{16}$$

**Notation 2. Activation:** Let the input activation matrix

$$\boldsymbol{X} = [\mathbf{x}_1, \mathbf{x}_2, \cdots, \mathbf{x}_b]^T \in \mathbb{R}^{b \times m} \tag{17}$$

,has mean $\nu$ and variance $\tau$ an in each row $\mathbf{x}_i$ has mean and variance:

$$\nu_i = \frac{1}{b} \sum_{k=1}^{b} x_{ki}, \quad \tau_i^2 = \frac{1}{b} \sum_{k=1}^{b} (x_{ki} - \nu_i)^2 \tag{18}$$

and note $\mathbf{A} = \mathbf{X}^\top \mathbf{X} \in \mathbb{R}^{m \times m}$.

**Notation 3. Error form:** Denote the matrix after quantization to $\hat{W}$ has a error matrix:

$$\boldsymbol{E} = \boldsymbol{W} - \hat{\boldsymbol{W}} = [\boldsymbol{E}_1, \ldots, \boldsymbol{E}_n]. \tag{19}$$

, and the loss in quantization satisfies:

$$\mathcal{L} = \|\mathbf{X}\boldsymbol{E}\|_F^2 = \sum_{k=1}^{b} \|\mathbf{x}_k \boldsymbol{E}\|^2 = \sum_{k=1}^{b} \left\| \sum_{j=1}^{n} (\mathbf{x}_k \boldsymbol{E}_j) \right\|^2 = \sum_{j=1}^{n} \sum_{l=1}^{n} \boldsymbol{E}_j^\top (\mathbf{X}^\top \mathbf{X}) \boldsymbol{E}_l = \sum_{j=1}^{n} \sum_{l=1}^{n} \boldsymbol{E}_j^\top \mathbf{A} \boldsymbol{E}_l \tag{20}$$

where $\mathbf{x}_k$ is the $k$-th input sample (row vector).

**Notation 4. Hadamard Transformation:** Define the normalized Hadamard matrix $\mathbf{H} \in \mathbb{R}^{n \times n}$, satisfying $\mathbf{H}\mathbf{H}^\top = \mathbf{I}$, with elements in $[+\frac{1}{\sqrt{n}}, -\frac{1}{\sqrt{n}}]$. Denote rotated weights:

$$\mathbf{W}' = \mathbf{W}\mathbf{H} = [\mathbf{w}_1', \mathbf{w}_2', \ldots, \mathbf{w}_n'] \tag{21}$$

where:

$$\mathbf{w}_k' = \sum_{j=1}^{n} h_{jk} \mathbf{w}_j \tag{22}$$

After quantization and inverse rotation:

$$\hat{\boldsymbol{W}} = \hat{\boldsymbol{W}}' \mathbf{H}^\top, \quad \boldsymbol{E} = \boldsymbol{E}' \mathbf{H}^\top \tag{23}$$

**Assumptions 1.** Let input matrix $\mathbf{X}$ Under the following assumption:

- Input means $\nu_i$ are small or centered (so that $\tau_i^2 \approx \frac{1}{b} \sum_{k=1}^{b} x_{ki}^2$).

**Assumptions 2.** Assume that the quantization error $\boldsymbol{E}_j$ is proportional to the centered weight vector:

$$e_{ij} \propto (w_{ij} - \mu_j) \cdot \varepsilon_j \tag{24}$$

where $\varepsilon_j$ is the unit quantization error factor for column $j$.

B.2 PROOF 1. QUANTIZATION UNDER ROTATION

**Theorem 1.** *Denote $\mathcal{L}_{orig}$ as the origin quantization loss, and $\mathcal{L}_{rot}$ as the loss under rotation; we can deduce that,*

$$\mathcal{L}_{\text{rot}}(\boldsymbol{R}) \leq \mathcal{L}_{\text{orig}}(\boldsymbol{R}) \tag{25}$$

*Proof.*

*i).* **Error before rotation:**

From Equation equation 20,

$$\begin{aligned}
\boldsymbol{E}_j^\top \mathbf{A} \boldsymbol{E}_l &= \sum_{i=1}^{m} \sum_{p=1}^{m} e_{ij} e_{pl} A_{ip} \\
&= \underbrace{\sum_{i=1}^{m} e_{ij} e_{il} A_{ii}}_{\text{diagonal terms}} + \underbrace{\sum_{i \neq p} e_{ij} e_{pl} A_{ip}}_{\text{cross terms } c} \\
&\geq b \cdot \sum_{i=1}^{m} \tau_i^2 e_{ij} e_{il}
\end{aligned} \tag{26}$$

Define the weighted inner product and apply assumptions in Eq 24:

$$\langle \boldsymbol{E}_j, \boldsymbol{E}_l \rangle_{\boldsymbol{\tau}} = \sum_{i=1}^{m} \tau_i^2 e_{ij} e_{il}$$

$$\propto \varepsilon_j \varepsilon_l \sum_{i=1}^{m} \tau_i^2 (w_{ij} - \mu_j)(w_{il} - \mu_l) \tag{27}$$

Define the weighted covariance:

$$\mathrm{Cov}_{\boldsymbol{\tau}}(j, l) = \sum_{i=1}^{m} \tau_i^2 (w_{ij} - \mu_j)(w_{il} - \mu_l) \tag{28}$$

and the weighted variance:

$$\mathrm{Var}_{\boldsymbol{\tau}}(j) = \mathrm{Cov}_{\boldsymbol{\tau}}(j, j) = \sum_{i=1}^{m} \tau_i^2 (w_{ij} - \mu_j)^2 \tag{29}$$

For loss in Eq 20:

$$\mathcal{L}_{\mathrm{orig}} \geq b \cdot \sum_{j=l}^{n} \sum_{l=1}^{n} \langle \boldsymbol{E}_j, \boldsymbol{E}_l \rangle_{\boldsymbol{\tau}}$$

$$\propto b \cdot \sum_{j=1}^{n} \sum_{l=1}^{n} \varepsilon_j \varepsilon_l \cdot \mathrm{Cov}_{\boldsymbol{\tau}}(j, l) \tag{30}$$

$$= \sum_{j=1}^{n} \varepsilon_j^2 \cdot \mathrm{Var}_{\boldsymbol{\tau}}(j) + \sum_{j \neq l} \varepsilon_j \varepsilon_l \cdot \mathrm{Cov}_{\boldsymbol{\tau}}(j, l)$$

***ii).Error after rotation:***

After rotation: $\mathbf{W}' = \mathbf{W}\mathbf{H}$, $\boldsymbol{E} = \boldsymbol{E}'\mathbf{H}^{\top}$.

Let $\boldsymbol{E}'_k$ be the $k$-th column of $\boldsymbol{E}'$. Then:

$$\boldsymbol{E}_j = \sum_{k=1}^{n} h_{kj} \boldsymbol{E}'_k \tag{31}$$

Substitute into the loss:

$$\begin{aligned}
\mathcal{L}_{\mathrm{rot}} &= \|\mathbf{X}\boldsymbol{E}\|_F^2 = \|\mathbf{X}\boldsymbol{E}'\mathbf{H}^{\top}\|_F^2 \\
&= \mathrm{Tr}\left((\boldsymbol{E}'\mathbf{H}^{\top})^{\top}\mathbf{X}^{\top}\mathbf{X}(\boldsymbol{E}'\mathbf{H}^{\top})\right) \\
&= \mathrm{Tr}\left(\mathbf{H}\boldsymbol{E}'^{\top}\mathbf{A}\boldsymbol{E}'\mathbf{H}^{\top}\right) \\
&= \mathrm{Tr}\left(\boldsymbol{E}'^{\top}\mathbf{A}\boldsymbol{E}'\mathbf{H}^{\top}\mathbf{H}\right) \\
&= \mathrm{Tr}\left(\boldsymbol{E}'^{\top}\mathbf{A}\boldsymbol{E}'\right) \\
&= \sum_{k=1}^{n} \boldsymbol{E}'^{\top}_k \mathbf{A}\boldsymbol{E}'_k
\end{aligned} \tag{32}$$

Since $\sum_{j=1}^{n} h_{kj} h_{pj} = \delta_{kp}$ (due to orthogonality of $\mathbf{H}$), cross-column terms can be eliminated.

Thus:

$$\mathcal{L}_{\mathrm{rot}} = b \cdot \sum_{k=1}^{n} \langle \boldsymbol{E}'_k, \boldsymbol{E}'_k \rangle_{\boldsymbol{\tau}} = b \cdot \sum_{k=1}^{n} \mathrm{Var}'_{\boldsymbol{\tau}}(k) \cdot (\varepsilon'_k)^2$$

$$\propto \sum_{k=1}^{n} (\varepsilon'_k)^2 \cdot \mathrm{Var}'_{\boldsymbol{\tau}}(k) \tag{33}$$

where $\mathrm{Var}'_{\boldsymbol{\tau}}(k) = \sum_{i=1}^{m} \tau_i^2 (w'_{ik} - \mu'_k)^2$ is the weighted variance of the $k$-th rotated column, and $\varepsilon'_k$ is the unit quantization error in the rotated space.

***iii).*Comprehensive analysis:** Analyze the rotated weighted variance:

$$
\begin{aligned}
\mathrm{Var}'_{\boldsymbol{\tau}}(k) &= \sum_{i=1}^{m} \tau_i^2 \left( \sum_{j=1}^{n} h_{jk}(w_{ij} - \mu_j) \right)^2 \\
&= \sum_{j=1}^{n} h_{jk}^2 \underbrace{\sum_{i=1}^{m} \tau_i^2 (w_{ij} - \mu_j)^2}_{\mathrm{Var}_{\boldsymbol{\tau}}(j)} + \sum_{j_1 \neq j_2} h_{j_1 k} h_{j_2 k} \underbrace{\sum_{i=1}^{m} \tau_i^2 (w_{ij_1} - \mu_{j_1})(w_{ij_2} - \mu_{j_2})}_{\mathrm{Cov}_{\boldsymbol{\tau}}(j_1, j_2)}
\end{aligned}
\tag{34}
$$

Since $h_{jk}^2 = \frac{1}{n}$, the first term is:

$$
\frac{1}{n} \sum_{j=1}^{n} \mathrm{Var}_{\boldsymbol{\tau}}(j) = \overline{\mathrm{Var}_{\boldsymbol{\tau}}}
\tag{35}
$$

The second term (cross-covariance) is approximately zero in practice due to sign oscillations in the Hadamard matrix. Thus:

$$
\mathrm{Var}'_{\boldsymbol{\tau}}(k) \approx \overline{\mathrm{Var}_{\boldsymbol{\tau}}} = \frac{1}{n} \sum_{j=1}^{n} \mathrm{Var}_{\boldsymbol{\tau}}(j)
\tag{36}
$$

In the rotated space, due to decorrelation, $\varepsilon'_k$ is stable and minimized, whereas in the original space, $\varepsilon_j$ is amplified by error propagation. Thus, statistically, $\varepsilon'_k \leq \varepsilon_j$.

Recall Eq 30, before rotation:

$$
\begin{aligned}
\mathcal{L}_{\mathrm{orig}} &\propto \sum_{j=1}^{n} \varepsilon_j^2 \cdot \mathrm{Var}_{\boldsymbol{\tau}}(j) + \underbrace{\sum_{j \neq l} \varepsilon_j \varepsilon_l \cdot \mathrm{Cov}_{\boldsymbol{\tau}}(j, l)}_{\geq 0 (\text{if positive correlation})} \\
&\geq \sum_{j=1}^{n} \varepsilon_j^2 \cdot \mathrm{Var}_{\boldsymbol{\tau}}(j) \\
&\geq n \cdot \varepsilon_{\min}^2 \cdot \overline{\mathrm{Var}_{\boldsymbol{\tau}}}
\end{aligned}
\tag{37}
$$

After rotation:

$$
\mathcal{L}_{\mathrm{rot}} \propto \sum_{k=1}^{n} (\varepsilon'_k)^2 \cdot \overline{\mathrm{Var}_{\boldsymbol{\tau}}} \leq n \cdot \varepsilon_{\min}^2 \cdot \overline{\mathrm{Var}_{\boldsymbol{\tau}}}
\tag{38}
$$

Therefore, we can prove:

$$
\mathcal{L}_{\mathrm{rot}} \leq n \cdot \varepsilon_{\min}^2 \cdot \overline{\mathrm{Var}_{\boldsymbol{\tau}}} \leq \mathcal{L}_{\mathrm{orig}}
\tag{39}
$$

Equality holds only if:

1. All $\varepsilon_j = \varepsilon_{\min}$ (no error propagation).
2. All $\mathrm{Cov}_{\boldsymbol{\tau}}(j, l) = 0$ (columns uncorrelated).
3. All $\mathrm{Var}_{\boldsymbol{\tau}}(j)$ are equal (variance balanced).

In practice, these conditions are rarely met, so typically $\mathcal{L}_{\mathrm{rot}} < \mathcal{L}_{\mathrm{orig}}$.

In LoPRo, quantization is applied to the partially rotated matrix $\boldsymbol{R}'$. Specifically, the first $b_I$ columns of $\boldsymbol{R}'$ are preserved unchanged to maintain the most significant components, while the remaining columns are partitioned into blocks of size $b_H \times b_H$ and transformed by $\boldsymbol{H}_{wal} \in \mathbb{R}^{b_H \times b_H}$ and satisfy $\mathcal{L}_{\mathrm{rot}} < \mathcal{L}_{\mathrm{orig}}$. Therefore, the theoretical error bound in Eq 39 still holds for the whole quantization. This ensures that LoPRo maintains the same global error minimization objective while enhancing local quantization stability through structured rotation.

### B.3 Proof 2. Quantization with hessian after rotation

**Theorem 2.** *Under a rotation Hessian-optimized quantization, we have*

$$\mathcal{L}(\boldsymbol{R}) = \|\boldsymbol{R}\boldsymbol{X} - \hat{\boldsymbol{R}}\boldsymbol{X}\|^2 = \mathrm{tr}\left((\hat{\boldsymbol{R}} - \boldsymbol{R})\boldsymbol{H}'(\hat{\boldsymbol{R}} - \boldsymbol{R})^T\right), \tag{40}$$

where $\boldsymbol{H}' = \boldsymbol{Q}^T \boldsymbol{P}^T \boldsymbol{H} \boldsymbol{P}$

*Proof.*

Given the form:

$$\boldsymbol{W} = \boldsymbol{W}_r + \boldsymbol{R}(\boldsymbol{Q}^\top \boldsymbol{Q}), \tag{41}$$

where $\boldsymbol{Q} = \boldsymbol{P}\boldsymbol{H}_{wal}$ is an orthogonal matrix from LoPRo (Eq 6) satisfying $\boldsymbol{Q}^\top \boldsymbol{Q} = \boldsymbol{I}$.

Now apply the $\ell$ of proxy Hessian by minimizing:

$$\mathcal{L}(\boldsymbol{W}) = \boldsymbol{E}_X \|(\boldsymbol{W}\boldsymbol{X} - (\boldsymbol{W}_r\boldsymbol{X} + \mathcal{Q}(\boldsymbol{R}\boldsymbol{Q}^T)\boldsymbol{Q})\boldsymbol{X}\| = \boldsymbol{E}_{\boldsymbol{X}} \|\boldsymbol{R}\boldsymbol{X} - \mathcal{Q}(\boldsymbol{R}\boldsymbol{Q}^T)\boldsymbol{Q}\boldsymbol{X}\| \tag{42}$$

Let $\tilde{\boldsymbol{X}} = \boldsymbol{Q}\boldsymbol{X}$ and $\boldsymbol{R}' = \boldsymbol{R}\boldsymbol{Q}^T$, then:

$$\mathcal{L}(\boldsymbol{W}) = \|\boldsymbol{R}\boldsymbol{Q}^T\tilde{\boldsymbol{X}} - \mathcal{Q}(\boldsymbol{R}\boldsymbol{Q}^T)\tilde{\boldsymbol{X}}\| \tag{43}$$

$$= \mathrm{tr}\left((\hat{\boldsymbol{R}'} - \boldsymbol{R})\boldsymbol{H}'(\hat{\boldsymbol{R}'} - \boldsymbol{R})^\top\right), \tag{44}$$

In this case: $\boldsymbol{H}' = \boldsymbol{Q}^T \boldsymbol{H} \boldsymbol{Q}$, take $\boldsymbol{Q} = \boldsymbol{P}\boldsymbol{H}_{wal}$, we can prove the Eq 9.

## C   Additional Implementation Details

**Set up:** Experiments with the MoE model Mixtral-8x7B were conducted on a single NVIDIA A800 80GB GPU, while all other experiments were performed on a single NVIDIA A100 40GB GPU. The difference in hardware only leads to minor variations in quantization costs and inference latency; all accuracy metrics and other numerical results remain identical across platforms, ensuring fair and consistent evaluation.

**Calibration:** We use a calibration dataset consisting of 128 randomly sampled sequences, each containing 2048 tokens, from c4 Raffel et al. (2020), a sampling strategy shown to be effective in OmniQuant Shao et al. (2023) and AffineQuant Ma et al. (2024).

**Evaluation:** For the perplexity (PPL) evaluation, we set the context length to match the maximum sequence length used during model training: 4096 for LLaMA-2 and 8192 for LLaMA-3. In zero-shot evaluations, we report the `acc` metric (rather than `acc_norm`) from the `lm-eval-harness` Gao et al. (2024). All results are rounded to one or two decimal places as appropriate. Here are brief introductions to the zero-shot datasets:

- **ARC-Challenge (AC)** and **ARC-Easy (AE)** Boratko et al. (2018): The AI2 Reasoning Challenge (ARC) dataset consists of multiple-choice science questions from grade school level. ARC-Challenge contains questions that are difficult for both retrieval and word co-occurrence methods, focusing on genuine reasoning, while ARC-Easy includes questions that are more amenable to simpler methods.
- **PIQA (QA)** Bisk et al. (2020): The Physical Interaction: Question Answering dataset evaluates a model's ability to understand physical commonsense reasoning. It presents questions about the physical properties and interactions of everyday objects, requiring models to choose the most plausible solution between two options.
- **Winogrande (WI)** Sakaguchi et al. (2021): Winogrande is a large-scale dataset designed to test commonsense reasoning, specifically tackling the Winograd Schema Challenge. It features a new adversarial filtering approach to create difficult multiple-choice questions that require understanding of context and pronoun resolution.
- **BoolQ (BQ)** Clark et al. (2019): The Boolean Questions dataset contains yes/no questions derived from real search queries paired with paragraphs from Wikipedia. The task is to determine the correct boolean answer based on the information in the given passage.

- **Hellaswag (HS)**Zellers et al. (2019): The HELLA SWAG dataset evaluates commonsense inference in sentence completion tasks. Given a partial sentence describing a situation, models must select the most plausible continuation from multiple choices, with adversarially generated distractors making the task challenging.

- **OpenbookQA (OB)**Mihaylov et al. (2018): OpenBookQA is a multiple-choice question-answering dataset that uses a "fact" from an open book as a basis for questions requiring both the provided fact and general knowledge to answer, aiming to test deeper understanding and reasoning.

- **MathQA (MQ)**Amini et al. (2019): MathQA is a dataset of math word problems paired with annotated solutions in a Python-like programming language. It is designed to evaluate and improve the ability of models to perform multi-step mathematical reasoning and solve quantitative problems.

**Implementation:** In the main pipeline of LoPRo, during the R1SVD low-rank approximation phase, we set the rank size $r$ to 16 and perform 8 iterations $it$. The matrices $U$ and $V$ are stored in the 8-bit floating-point format *e4m3*, as they are orthogonal matrices with a narrow dynamic range in $[-1,1]$; compared to *e5m2*, *e4m3* provides higher precision for such distributions. In the partial rotation phase under rearrangement, the block sizes for the identity matrix and the rotation matrix, denoted as $b_I$ and $b_H$, are both set to 256. A detailed ablation study of these hyperparameters and strategies is presented in Appendix D. In the implementation of the quantize residual matrix, for vector-level quantization, we do not employ advanced techniques such as block-wise scaling or learnable codebooks. Instead, we adopt the simplest form: a randomly initialized codebook combined with quantization based on Eq 9. Furthermore, we use $4D$ codebook for 2-bit quantization and $2D$ codebook for 3-bit quantization, following the optimal configuration used in GPTVQ. However, in comparison to our quantization scheme in LoPRo, for the GPTVQ baseline in main evaluation, we respect the integrity of the original method and only distinguishing whether fine-tuning (FT) is applied while adopting all other techniques (e.g., learnable codebooks, block-wise scaling) as specified in the original paper to ensure fair comparison using their best-reported configurations.

**Baseline:** Following the taxonomy introduced in § 2, we select representative and state-of-the-art baselines from each strategy category (Tag in ▮▮ ▮) to ensure comprehensive and fair comparisons. Our selection principle is twofold: (1) choose the strongest-performing method within each category, and (2) ensure full coverage of all major strategy tags. Specifically, for rotation-based quantization, we select QuIP#Tseng et al. (2024) — with the stronger performance than rotation methods like QuaRot Ashkboos et al. (2024), SpinQuant Liu et al. (2025), and DuQuant Lin et al. (2024a), which underperform QuIP# at sub-3-bit settings. For low-rank fine-tuning assisted quantization, we include RILQ Lee et al. (2025) and Caldera Saha et al. (2024), which represent the most competitive results in this category. For low-rank compensation quantization, we select LQER Zhang et al. (2024a) as both methods leverage low-rank components to recover quantization error. For loss and clipping method, we choose OminiQuant Shao et al. (2023). Finally , MoEQuant Chen et al. (2025) is choosen as the baseline in MoE model quantization. Additionally, we include GPTQ Frantar et al. (2023) and GPTVQ Van Baalen et al. (2024), which are leveraged within LoPRo to enable ablation and component-wise analysis. The multidimensional baseline selection ensures that LoPRo is evaluated against the strongest existing approaches and providing a holistic assessment of its effectiveness.

**Fine tuning:** We adopt CALDERA and RILQ, two state-of-the-art LoRA fine-tuning PTQ methods as baselines, and apply RILQ as the backend for fine-tuning in our LoPRo. In Table 2, both compared methods are implemented with their optimal configurations as reported in the original papers, specifically using QuIP# as the pre-quantization method prior to fine-tuning.

# D EXTENDED ABLATION STUDIES

**Overall:** In addition to the ablation on rotation strategy in §4.5, we conducted a comprehensive set of ablation studies to validate the effectiveness of key components and the hyperparameter chosen in LoPRo. Based on empirical evidence from these experiments, we finalize the following hyperparameter settings:

- **Low-Rank Approximation Method:** R1SVD

> – Low-rank approximation rank: $r = 16$
> – Number of R1SVD iterations: $iter = 8$

- **Rotation Block Parameters:** $b_I = b_H = 256$
- **Calibration Dataset:** 128 randomly sampled sequences from c4 dataset.

These configurations are supported by ablation results across multiple models and tasks, demonstrating robustness and consistent performance. Unless otherwise specified, all experiments in this work use the above settings to ensure fairness and reproducibility. The final strategies adopted in the experiment are marked with dark colors.

## D.1 ABLATION ON RANK

In LoPRo, we performed an ablation study on the choice of rank size, with results shown in Table 7. The results indicate that quantization accuracy generally improves as the rank increases. However, under vector quantization and 3-bit scalar quantization, the accuracy gains from increasing rank are marginal, suggesting that even small ranks are sufficient to capture the dominant structure in these settings.

Using an excessively large rank yields diminishing returns in accuracy while significantly reducing the compression ratio and increasing memory footprint. Therefore, for models of various sizes, we find that a rank of 16 strikes an optimal balance: it achieves high quantization accuracy with minimal overhead—adding less than 3% extra memory cost—while maintaining fast decomposition and inference. This overhead further decreases as model scale increases, making $r = 16$ a practical and scalable choice across architectures.

## D.2 ABLATION ON LOW-RANK DECOMPOSITION

We conduct an ablation study on R1SVD, with results presented in Table 8. Due to fluctuations in GPU computational performance, the reported execution times exhibit minor variability. Nevertheless, the results clearly show that the runtime of R1SVD scales nearly linearly with model size—requiring only about 1 minute for the 7B model and approximately 2 minutes for the 13B model.

The computational cost of R1SVD is dominated by *iter* GEMV (General Matrix-Vector Multiplication) operations per weight matrix, which can be approximated as equivalent to performing *iter* forward passes with batch size 1. This makes it highly efficient and scalable.

In contrast, SVD-based decomposition is significantly slower: it takes 14 minutes for the 7B model and over 30 minutes for the 13B model. Moreover, SVD suffers from poor GPU parallelization, and most standard libraries (e.g., cuSOLVER) do not support *fp16* computation. As a result, SVD runs exclusively in *fp32* or higher precision, further increasing its computational burden. Critically, R1SVD maintains high numerical accuracy despite using mixed *fp16/fp8* arithmetic. This is because the precision loss from each *fp16*-to-*fp8* conversion is compensated in subsequent iterations, akin to the error feedback mechanism in OBS (Optimal Brain Surgeon). Consequently, the accuracy of R1SVD closely approaches that of full-precision SVD.

In summary, the R1SVD achieves approximation quality comparable to that of SVD while being orders of magnitude faster. Given the relative tolerance to numerical error in deep learning compared to scientific computing, we consider that R1SVD holds strong potential for broad application in efficient model compression and large-scale training.

## D.3 ABLATION ON ITERATION

In the R1SVD algorithm, the sketch computation involves iteratively applying the $S^* A^* (AA^*)^{it}$ operation in Eq 12 for *it* times (corresponding to Line 13 in Algorithm 1). A higher iteration count improves approximation accuracy but incurs additional computational cost.

To evaluate the impact of this parameter, we present the performance of LoPRo under different iteration values in Table 9. The results show that quantization accuracy improves with increasing *it* and eventually plateaus. For the LLaMA-2 7B model, the accuracy stabilizes when $it \geq 8$, with perplexity (PPL) fluctuating by less than 0.01 — indicating diminishing returns beyond this point.

Table 7: Ablation study on rank selection in LoPRo. PPL (wiki2,ctx=4096) and Accuracy are measured uder different settings. 'r.bit' denote for the average bit of low-rank component. Abbreviations for zero-shot tasks follow those defined in §4.1 and 'Avg' stands for the average accuracy of four tasks.

| model | bit | method | rank | r.bit | PPL | AC | AE | QA | WI | Avg |
|---|---|---|---|---|---|---|---|---|---|---|
| LLaMA2-7B | 2 | LoPRo | 8 | 0.02 | 7.51 | 29.4 | 63.9 | 70.6 | 66.8 | 57.7 |
| | | | 16 | 0.05 | 7.39 | 31.2 | 62.8 | 71.1 | 63.8 | 57.2 |
| | | | 32 | 0.10 | 7.35 | 29.9 | 64.0 | 71.2 | 63.7 | 57.2 |
| | | | 64 | 0.19 | 7.30 | 31.8 | 65.2 | 70.8 | 64.3 | 58.0 |
| | | LoPRo$_v$ | 8 | 0.02 | 6.56 | 33.6 | 69.4 | 73.0 | 65.7 | 60.4 |
| | | | 16 | 0.05 | 6.54 | 34.6 | 69.0 | 72.7 | 66.5 | 60.7 |
| | | | 32 | 0.10 | 6.54 | 34.9 | 69.5 | 73.6 | 65.8 | 60.9 |
| | | | 64 | 0.19 | 6.49 | 34.0 | 69.5 | 73.1 | 66.2 | 60.7 |
| | 3 | LoPRo | 8 | 0.02 | 5.44 | 41.3 | 74.8 | 76.7 | 68.4 | 65.3 |
| | | | 16 | 0.05 | 5.43 | 41.0 | 74.9 | 76.3 | 68.9 | 65.3 |
| | | | 32 | 0.10 | 5.42 | 40.6 | 74.4 | 76.8 | 68.4 | 65.1 |
| | | | 64 | 0.19 | 5.41 | 41.0 | 75.0 | 77.3 | 69.1 | 65.6 |
| | | LoPRo$_v$ | 8 | 0.02 | 5.46 | 42.0 | 75.3 | 77.5 | 67.1 | 65.5 |
| | | | 16 | 0.05 | 5.45 | 41.0 | 74.8 | 76.7 | 69.1 | 65.4 |
| | | | 32 | 0.10 | 5.45 | 41.6 | 75.0 | 77.0 | 68.9 | 65.6 |
| | | | 64 | 0.19 | 5.44 | 39.7 | 74.5 | 77.3 | 68.0 | 64.9 |
| LLaMA2-13B | 2 | LoPRo | 8 | 0.02 | 6.49 | 33.3 | 66.9 | 71.2 | 65.5 | 59.2 |
| | | | 16 | 0.04 | 6.48 | 33.6 | 69.0 | 72.4 | 66.3 | 60.3 |
| | | | 32 | 0.08 | 6.40 | 33.9 | 68.6 | 72.9 | 66.1 | 60.4 |
| | | | 64 | 0.15 | 6.40 | 35.3 | 69.4 | 74.3 | 67.9 | 61.7 |
| | | LoPRo$_v$ | 8 | 0.02 | 5.87 | 38.3 | 73.1 | 74.4 | 68.7 | 63.6 |
| | | | 16 | 0.04 | 5.79 | 38.8 | 74.2 | 75.4 | 68.0 | 64.1 |
| | | | 32 | 0.08 | 5.76 | 38.3 | 73.7 | 74.6 | 67.6 | 63.6 |
| | | | 64 | 0.15 | 5.71 | 38.6 | 73.1 | 75.6 | 68.2 | 63.9 |
| | 3 | LoPRo | 8 | 0.02 | 4.85 | 46.1 | 78.0 | 78.0 | 70.4 | 68.1 |
| | | | 16 | 0.04 | 4.84 | 44.9 | 78.7 | 78.5 | 71.1 | 68.3 |
| | | | 32 | 0.08 | 4.84 | 44.1 | 77.3 | 78.0 | 72.4 | 67.9 |
| | | | 64 | 0.15 | 4.81 | 46.2 | 78.2 | 78.5 | 72.9 | 68.9 |
| | | LoPRo$_v$ | 8 | 0.02 | 4.91 | 43.3 | 76.8 | 77.6 | 71.9 | 67.4 |
| | | | 16 | 0.04 | 4.87 | 44.5 | 77.4 | 78.5 | 71.0 | 67.9 |
| | | | 32 | 0.08 | 4.86 | 44.8 | 77.3 | 78.2 | 70.0 | 67.6 |
| | | | 64 | 0.15 | 4.85 | 44.6 | 77.9 | 78.2 | 70.2 | 67.7 |

Furthermore, although each additional iteration increases the low-rank approximation time, this stage constitutes only a small fraction of the overall quantization pipeline. For instance, even with 16 iterations, the R1SVD step takes less than one minute, making it highly efficient in practice. Therefore, we set $it = 8$ as the default in all other experiments, achieving near-optimal accuracy while maintaining high computational efficiency and scalability.

### D.4 ABLATION ON BLOCK SIZE

In LoPRo's rotation stage, we introduce two block size parameters: $b_I$ and $b_H$, which respectively denote the block size of the identity matrix in the upper matrix and the block size of the Hadamard-Walsh (Hwal) matrix in the lower matrix of the partial block rotation matrix.

We evaluate the impact of different block configurations on the LLaMA-2 7B model, with results presented in Table 10. Since $Hwal$ block size must satisfy $b_H \in \mathbb{R}^{2^i}$, and the second dimension of weight matrices (e.g., in MLP layers) is typically a multiple of 128, we restrict our test cases to $256 > b_I \geq b_H > 64$. This constraint ensures that: (1) $b_H$ remains a power of two, (2) the total number of blocks is an integer, and (3) since the MLP layer dimensions in LLaMA-2 7B are not divisible by $b_H$ when $i > 9$ ($b_H = 512$). We further enforce $b_I > b_H$ and multiples of 64 to guarantee valid integer partitioning.

Table 8: Ablation study on low-rank method and bit in LoPRo. 'LoBit' stand for the precision of $U, V$ in low-rank matrix. Time$_{tot}$ and Time$_{low}$ respectively represent the total execution time of the algorithm and the execution time of the low-rank approximation.

| Model | Method | LoBit | LoRA | Time$_{tot}$ | Time$_{low}$ | PPL | AC | AE | QA | WI |
|---|---|---|---|---|---|---|---|---|---|---|
| LLaMA2-7B | LoPRo | 8 | R1SVD | 26.4m | 0.8m | 7.39 | 31.2 | 62.8 | 71.1 | 63.8 |
| | | | SVD | 42.2m | 14.0m | 7.44 | 31.1 | 62.0 | 69.6 | 64.6 |
| | | 16 | R1SVD | 27.3m | 1.2m | 7.38 | 30.0 | 64.1 | 71.5 | 62.4 |
| | | | SVD | 42.3m | 14.0m | 7.40 | 31.4 | 62.2 | 70.0 | 64.5 |
| | LoPRo$_v$ | 8 | R1SVD | 31.7m | 1.1m | 6.53 | 34.6 | 69.0 | 72.7 | 66.5 |
| | | | SVD | 47.2m | 14.2m | 6.52 | 35.0 | 67.9 | 73.2 | 66.0 |
| | | 16 | R1SVD | 31.5m | 1.1m | 6.55 | 35.5 | 70.7 | 73.7 | 66.2 |
| | | | SVD | 47.2m | 14.0m | 6.56 | 34.8 | 70.5 | 73.9 | 64.7 |
| LLaMA2-13B | LoPRo | 8 | R1SVD | 45.1m | 2.2m | 6.48 | 33.6 | 69.0 | 72.4 | 66.3 |
| | | | SVD | 1.4h | 31.2m | 6.54 | 34.0 | 68.2 | 72.1 | 66.3 |
| | | 16 | R1SVD | 45.8m | 3.2m | 6.52 | 32.8 | 66.2 | 71.7 | 65.2 |
| | | | SVD | 1.4h | 31.4m | 6.48 | 33.5 | 69.3 | 72.5 | 64.6 |
| | LoPRo$_v$ | 8 | R1SVD | 56m | 2.1m | 5.79 | 38.8 | 74.2 | 75.4 | 68.0 |
| | | | SVD | 1.6h | 31.8m | 5.88 | 37.6 | 72.4 | 75.4 | 67.8 |
| | | 16 | R1SVD | 57m | 2.8m | 5.77 | 38.6 | 73.5 | 75.0 | 67.8 |
| | | | SVD | 1.6h | 31.7m | 5.75 | 38.8 | 73.5 | 74.9 | 67.9 |

Table 9: Ablation study on iteration $it$ (R1SVD - Eq 12) in LLaMA2-7B. Time$_{low}$ stands for the execution time of low-rank approximation.

| Method | Bit | Iteration | PPL | AC | AE | QA | WI | Time$_{low}$ |
|---|---|---|---|---|---|---|---|---|
| LoPRo | 2 | 1 | 7.44 | 31.0 | 64.9 | 70.0 | 64.1 | 0.1m |
| | | 2 | 7.51 | 29.8 | 63.9 | 69.9 | 64.0 | 0.2m |
| | | 4 | 7.41 | 30.3 | 64.4 | 70.6 | 64.1 | 0.4m |
| | | 8 | 7.39 | 31.2 | 62.8 | 71.1 | 63.8 | 0.8m |
| | | 16 | 7.40 | 31.3 | 62.0 | 71.1 | 63.5 | 1.6m |
| | | 32 | 7.40 | 31.3 | 62.1 | 71.6 | 64.1 | 3.2m |
| | 3 | 1 | 5.43 | 41.1 | 74.8 | 76.4 | 68.0 | 0.1m |
| | | 2 | 5.42 | 41.0 | 73.9 | 77.2 | 67.9 | 0.2m |
| | | 4 | 5.43 | 40.5 | 74.3 | 77.1 | 69.1 | 0.4m |
| | | 8 | 5.43 | 41.0 | 74.9 | 76.3 | 68.9 | 0.8m |
| | | 16 | 5.42 | 41.2 | 74.2 | 77.3 | 67.9 | 1.6m |
| | | 32 | 5.42 | 41.0 | 74.6 | 76.6 | 67.9 | 3.2m |
| LoPRo$_v$ | 2 | 1 | 6.58 | 34.5 | 70.5 | 73.5 | 66.7 | 0.1m |
| | | 2 | 6.56 | 34.0 | 68.7 | 73.7 | 64.6 | 0.2m |
| | | 4 | 6.54 | 34.0 | 70.8 | 73.7 | 65.2 | 0.4m |
| | | 8 | 6.53 | 34.6 | 69.0 | 72.7 | 66.5 | 0.8m |
| | | 16 | 6.55 | 34.2 | 70.8 | 73.1 | 65.8 | 1.6m |
| | | 32 | 6.53 | 34.0 | 70.8 | 73.7 | 65.9 | 3.2m |
| | 3 | 1 | 5.46 | 41.1 | 75.1 | 76.1 | 67.9 | 0.1m |
| | | 2 | 5.45 | 39.6 | 74.9 | 76.7 | 68.0 | 0.2m |
| | | 4 | 5.45 | 39.9 | 74.6 | 76.8 | 67.8 | 0.4m |
| | | 8 | 5.45 | 41.0 | 74.8 | 76.7 | 69.1 | 0.8m |
| | | 16 | 5.45 | 41.6 | 75.1 | 77.3 | 69.8 | 1.6m |
| | | 32 | 5.45 | 39.9 | 74.9 | 76.4 | 67.8 | 3.2m |

The results indicate that block size parameters have a measurable, though moderate, impact on quantization accuracy. Specifically, smaller block sizes (e.g., $b_I, b_H < 128$) lead to slightly degraded performance, while configurations with block sizes larger than 128 yield comparable accuracy — with $b_I = b_H = 256$ showing a marginal advantage.

This behavior can be attributed to two factors: (1) When $b_I$ is too small, the more important channels (typically concentrated in the leading segment after permutation) are subjected to excessive rotation, which disrupts their numerical structure and increases quantization error. (2) Smaller rotation blocks do not adequately balance the distribution within each block, as they cannot effectively smooth out the quantization error.

Therefore, for consistency and simplicity across all experiments, we fix $b_I = b_H = 256$ as the default configuration, ensuring reproducible and stable performance without sacrificing accuracy.

Table 10: Ablation on block size parameters in 2bit LLaMA2-7B quantization.

| Method | $b_I$ | $b_H$ | PPL | AC | AE | QA | WI |
|---|---|---|---|---|---|---|---|
| LoPRo | 64 | 64 | 7.51 | 32.9 | 64.1 | 70.2 | 64.5 |
| | 128 | 64 | 7.43 | 29.9 | 64.4 | 70.7 | 64.5 |
| | 128 | 128 | 7.46 | 31.6 | 64.3 | 70.2 | 64.9 |
| | 256 | 64 | 7.40 | 30.8 | 63.9 | 71.7 | 63.1 |
| | 256 | 128 | 7.38 | 32.1 | 62.8 | 71.2 | 64.0 |
| | 256 | 256 | 7.39 | 31.2 | 62.8 | 71.1 | 63.8 |
| LoPRo$_v$ | 64 | 64 | 6.60 | 34.1 | 69.8 | 73.2 | 65.9 |
| | 128 | 64 | 6.56 | 35.5 | 68.1 | 72.5 | 66.3 |
| | 128 | 128 | 6.53 | 34.0 | 69.4 | 72.9 | 65.4 |
| | 256 | 64 | 6.55 | 33.6 | 69.2 | 72.0 | 65.9 |
| | 256 | 128 | 6.53 | 34.8 | 69.0 | 72.5 | 65.5 |
| | 256 | 256 | 6.53 | 34.6 | 69.0 | 72.7 | 66.5 |

### D.5 ABLATION ON CALIBRATION DATASET

We evaluated the quantization performance of LoPRo in different calibration datasets, with results presented in Table 11. The results demonstrate that LoPRo maintains consistent performance advantages across WikiText-2, c4, Pile with minimal metric fluctuations. This indicates strong generalization capability across diverse domains and text styles, and confirms that LoPRo is not sensitive to the specific statistical properties of any single calibration set. This robustness aligns with the design philosophy of LoPRo as a fine-tuning-free quantization framework. Consequently, for all other experiments in this work, we adopt c4 as the default calibration dataset.

Table 11: Ablation on calibration dataset in 2bit LLaMA2-7B quantization..

| Method | Dataset | PPL | AC | AE | QA | WI |
|---|---|---|---|---|---|---|
| LoPRo | Wiki2 | 7.49 | 32.9 | 64.1 | 70.2 | 64.5 |
| | C4 | 7.39 | 31.2 | 62.8 | 71.1 | 63.8 |
| | Pile | 7.46 | 31.6 | 64.3 | 70.2 | 64.9 |
| LoPRo$_v$ | Wiki2 | 6.55 | 34.1 | 69.8 | 73.2 | 65.9 |
| | C4 | 6.53 | 34.6 | 69.0 | 72.7 | 66.5 |
| | Pile | 6.52 | 34.0 | 69.4 | 73.9 | 65.4 |

## E  MOE RESULTS

Since the MoEQuant Chen et al. (2025) paper neither reports results on several Zero-Shot benchmarks used in our main experiments nor provides open-source code, we introduce additional evaluation datasets — including BoolQ (BQ) Clark et al. (2019), Hellaswag (HS) Zellers et al. (2019), OpenbookQA (OB) Mihaylov et al. (2018), MathQA (MQ) Amini et al. (2019) to ensure a fair and comprehensive comparison. Results are summarized in Table 12.

**Comparison with MoEQuant:** LoPRo consistently outperforms MoEQuant across all evaluation tasks. Notably, under 2-bit quantization, LoPRo matches or exceeds the accuracy of MoEQuant at 3-bit precision — validating the trend observed in our main experiments. Specifically, on the BoolQ dataset, LoPRo achieves a +6 point improvement in accuracy; on other benchmarks, it performs comparably or better, while also exhibiting lower perplexity (i.e., reduced ambiguity).

This demonstrates that LoPRo can achieve *3-bit-level accuracy using only 2-bit weights* — a significant compression advantage. Moreover, quantizing the full 56B-parameter model (Mixtral-8x7B) takes only approximately 2.5 hours, highlighting the exceptional efficiency of our method. This combination of high accuracy, strong compression, and rapid quantization makes LoPRo particularly well-suited for deploying massive MoE models in resource-constrained environments.

Table 12: Performance of LoPRo and MoeQuant in Mixtral-8x7B model. Context length is 4096 and the abbreviations of zero-shot tasks are given in E.

| Method | Bit | PPL | AC | AE | WI | QA | HS | OB | BQ | MQ |
|--------|-----|-----|----|----|----|----|----|----|----|----|
| MoeQuant++ | 3 | 4.90 | - | - | - | - | 60.1 | 31.2 | 82.8 | 38.8 |
| LoPRo | 2.16 | 5.24 | 39.2 | 79.6 | 71.0 | 79.0 | 56.1 | 30.6 | 87.3 | 33.8 |
|  | 3.16 | 4.15 | 61.0 | 86.3 | 77.6 | 82.8 | 65.4 | 35.0 | 88.2 | 43.3 |
| LoPRo$_v$ | 2.16 | 4.80 | 55.9 | 83.8 | 73.4 | 80.4 | 59.4 | 32.2 | 87.3 | 37.5 |
|  | 3.16 | 4.15 | 60.8 | 86.1 | 76.9 | 83.5 | 65.0 | 36.4 | 87.7 | 42.9 |

# F    QWEN RESULTS

We evaluate LoPRo and its variant LoPRo$_v$ on three recent Qwen models — Qwen2.5-7B, Qwen2.5-14B, and Qwen3-8B — under both 2-bit and 3-bit weight quantization (with 16-bit activations). As shown in Table 13, LoPRo consistently preserves strong performance even at aggressive 2-bit compression. For Qwen2.5-7B, LoPRo$_v$ outperforms LoPRo by up to 4.5% in accuracy (e.g., on AC), demonstrating the benefit of variance-sensitive routing. At 3-bit precision, both LoPRo and LoPRo$_v$ nearly recover full FP16 performance across all benchmarks, with LoPRo$_v$ matching or exceeding FP16 on AE and QA for Qwen2.5-7B. The trend holds for larger models. In Qwen2.5-14B, LoPRo$_v$ at W2A16 reduces perplexity by 1.13 compared to LoPRo and achieves +2.8 higher accuracy on AC. Even in the more compact Qwen3-8B architecture, LoPRo$_v$ significantly narrows the gap to FP16: its W2A16 configuration attains 47.9 on AC versus 55.6 for FP16.

These results confirm that *LoPRo enables near-FP16 quality at 3-bit and usable performance at 2-bit across diverse Qwen architectures*. The consistent gains from LoPRo$_v$ further validate our design choice of incorporating activation variance into the routing mechanism. Combined with fast quantization runtime (empirically under 1 hour for all models on a single A100-40G), LoPRo offers a practical solution for deploying high-performance, compressed Qwen models in memory- and latency-constrained scenarios.

Table 13: Performance of LoPRo Qwen2 and Qwen3 model families.

| Models | Methods | Q Config | Wiki | AC | AE | WI | QA |
|--------|---------|----------|------|-----|-----|-----|-----|
| Qwen2.5-7b | Fp16 | W16A16 | 6.86 | 52.6 | 81.9 | 71.1 | 79.7 |
|  | LoPRo | W2A16 | 9.43 | 39.6 | 70.2 | 65.5 | 72.5 |
|  |  | W3A16 | 7.22 | 51.2 | 82.9 | 70 | 78.8 |
|  | LoPRo_v | W2A16 | 8.53 | 44.1 | 68.6 | 68 | 75.4 |
|  |  | W3A16 | 7.23 | 53.5 | 82.8 | 70.6 | 78.9 |
| Qwen2.5-14b | Fp16 | W16A16 | 5.24 | 60.7 | 85.7 | 75.6 | 81.6 |
|  | LoPRo | W2A16 | 7.75 | 47.4 | 76.7 | 71.9 | 75.7 |
|  |  | W3A16 | 5.44 | 57.8 | 84.4 | 75 | 79.4 |
|  | LoPRo_v | W2A16 | 6.62 | 50.2 | 78.8 | 72.5 | 77.3 |
|  |  | W3A16 | 5.42 | 57.2 | 83.9 | 74.7 | 70.8 |
| Qwen3-8b | Fp16 | W16A16 | 9.01 | 55.6 | 83.5 | 68.1 | 76.7 |
|  | LoPRo | W2A16 | 12.59 | 40.6 | 70.5 | 62.9 | 70.8 |
|  |  | W3A16 | 9.58 | 52.9 | 81.7 | 66.9 | 75.9 |
|  | LoPRo_v | W2A16 | 11.22 | 47.9 | 77.8 | 66.9 | 72.4 |
|  |  | W3A16 | 9.59 | 55.3 | 82.3 | 68.2 | 76 |

# G OPEN LLM LEADERBOARD V1

**Open LLM Leaderboard V1** provides a standardized evaluation suite for assessing language models on core academic benchmarks. It includes GSM8k Cobbe et al. (2021) for grade-school math reasoning, MMLU Hendrycks et al. (2020) and ARC-Challenge Boratko et al. (2018) for measuring world knowledge and logical reasoning, Winogrande Sakaguchi et al. (2021) and HellaSwag Zellers et al. (2019) for commonsense and language understanding, and TruthfulQA Lin et al. (2021) for evaluating factual correctness and resistance to generating false statements. Following Meta's prompt guidelines for Llama-3.1, the leaderboard treats MMLU and ARC-Challenge as text-generation tasks and applies chain-of-thought prompting to GSM8k, offering a consistent and reproducible protocol for comparing quantized and full-precision models.

We perform a OpenLLM V1 evaluation on Qwen3-8B, results show that both LoPRo and LoPRo_v achieve consistently strong performance across quantization settings. Even with aggressive 2-bit weight quantization (W2A16), the methods retain reasonable capability—particularly on tasks like Winograde—while W3A16 configurations recover over 94% of the FP16 baseline on average, demonstrating the effectiveness and robustness of the quantization approach across diverse benchmarks.

Table 14: Performance comparison of different quantization methods applied to the Qwen3-8B model across a suite of standard language modeling benchmarks. Recovery percentage is computed relative to the FP16 baseline (100% recovery). All evaluations follow the prompt and evaluation protocols aligned with the Open LLM Leaderboard V1, including chain-of-thought prompting for $\text{MMLU}_{Cot}$ and zero-shot settings for ARC-Challenge and TruthfulQA.

| Methods | Q Config | Recovery % | Average Score | MMLU 5-shot | $\text{MMLU}_{Cot}$ 0-shot | ARC-C 0-shot | GSM8K 8-shot | HellaSwag 10-shot | Winograde 5-shot | TruthfulQA 0-shot |
|---------|----------|-----------|---------------|-------------|------------------------------|--------------|--------------|--------------------|-------------------|-------------------|
| FP16 | - | 100.00% | 69.6 | 74.9 | 80.2 | 55.6 | 88.8 | 58.1 | 70.1 | 59.7 |
| LoPRo | W2A16 | 73.89% | 51.4 | 55.7 | 62.5 | 40.6 | 52.3 | 44.3 | 64.4 | 40.2 |
| | W3A16 | 94.40% | 65.7 | 71.1 | 75.3 | 54.3 | 81.2 | 53.9 | 69.9 | 54.2 |
| LoPRo_v | W2A16 | 79.89% | 55.6 | 61.8 | 67.9 | 47.9 | 55.5 | 47.2 | 66.2 | 42.7 |
| | W3A16 | 94.91% | 66.1 | 71.7 | 75.1 | 55.3 | 80.9 | 54.9 | 69.6 | 54.9 |

# H LIMITATIONS

This work primarily focuses on weight-only quantization. However, weight and activation quantization including KV-cache quantization — remains an active and critical area in model compression. We believe that the proposed rotation framework can be naturally extended to activation quantization, where structured rotation may similarly improve quantization efficiency by aligning activation distributions with quantization grids. Furthermore, the low-rank matrix multiplication and the reordering-rotation operations in LoPRo can be fused during inference, potentially enabling near-lossless computational efficiency with minimal overhead. We leave these directions for detailed exploration to future work.

# I BROADER IMPACTS

Our work identifies and addresses the coupling between different quantization strategies by analyzing the characteristics at each stage of the quantization pipeline, we show that a minimal low-rank component (e.g., rank = 16) is sufficient to capture the dominant information, enabling high-accuracy weight-only quantization without fine-tuning, this provides a natural starting point for subsequent LoRA-based fine-tuning. The pre-trained, high-precision $\mathbf{L}$ and $\mathbf{R}$ matrices can serve as initialization for LoRA adapters, potentially accelerating convergence and improving downstream task performance. Moreover, the proposed R1SVD algorithm offers a fast and scalable alternative to SVD, with near-optimal approximation quality and significantly lower computational cost ($\mathcal{O}(n^2)$ vs. $\mathcal{O}(n^3)$). Given the inherent robustness of large language models to small numerical perturbations, R1SVD is particularly well-suited for large-scale applications, where efficiency and scalability are paramount.

## J   LLMs USAGE

This paper presents a quantization algorithm tailored for LLMs, where the evaluation is conducted with LLMs weights. In addition, LLMs are solely employed for linguistic polishing.

## K   MORE VISUALIZATION

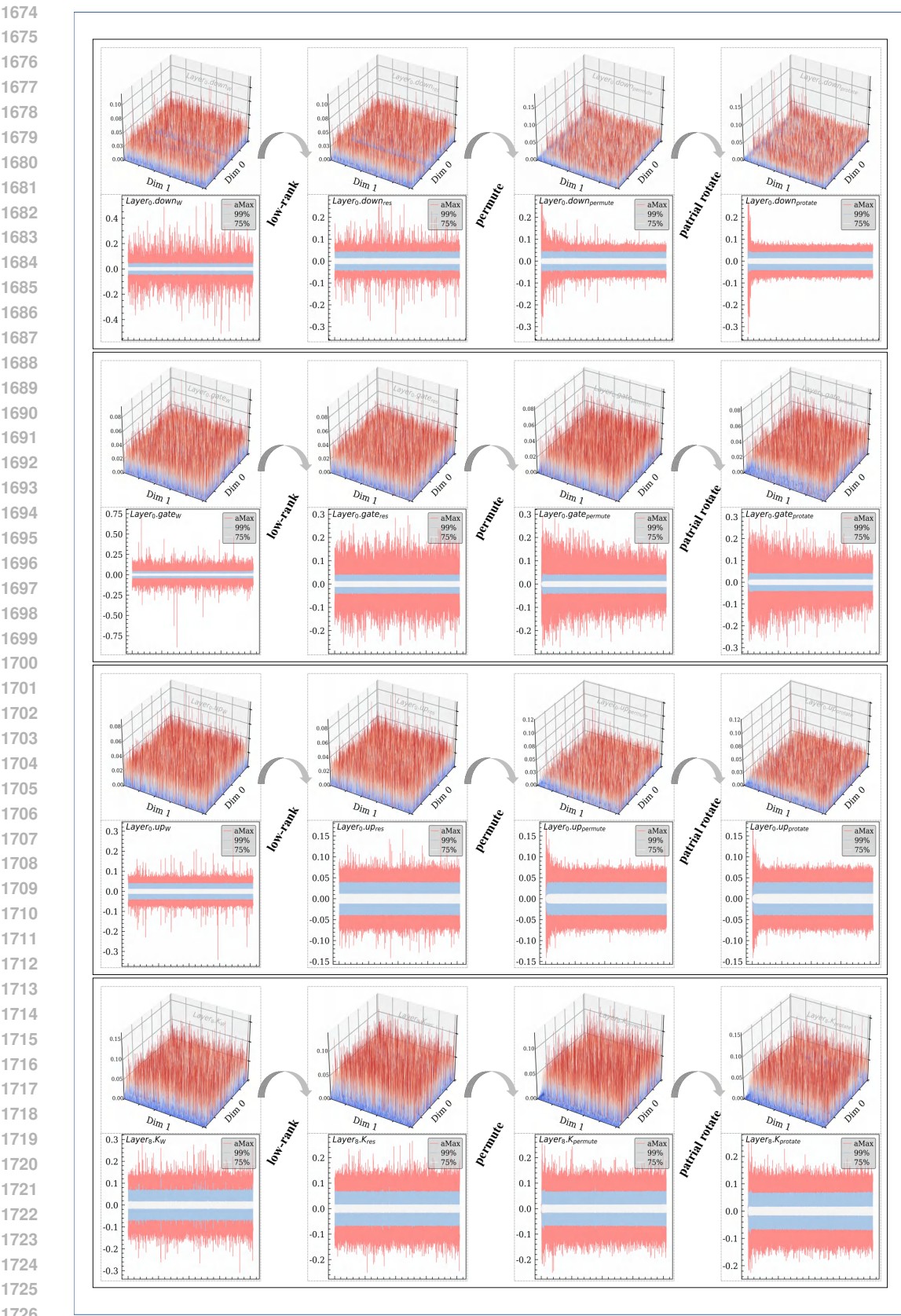

Figure 2: Visualization of layers in LLaMA2-7B

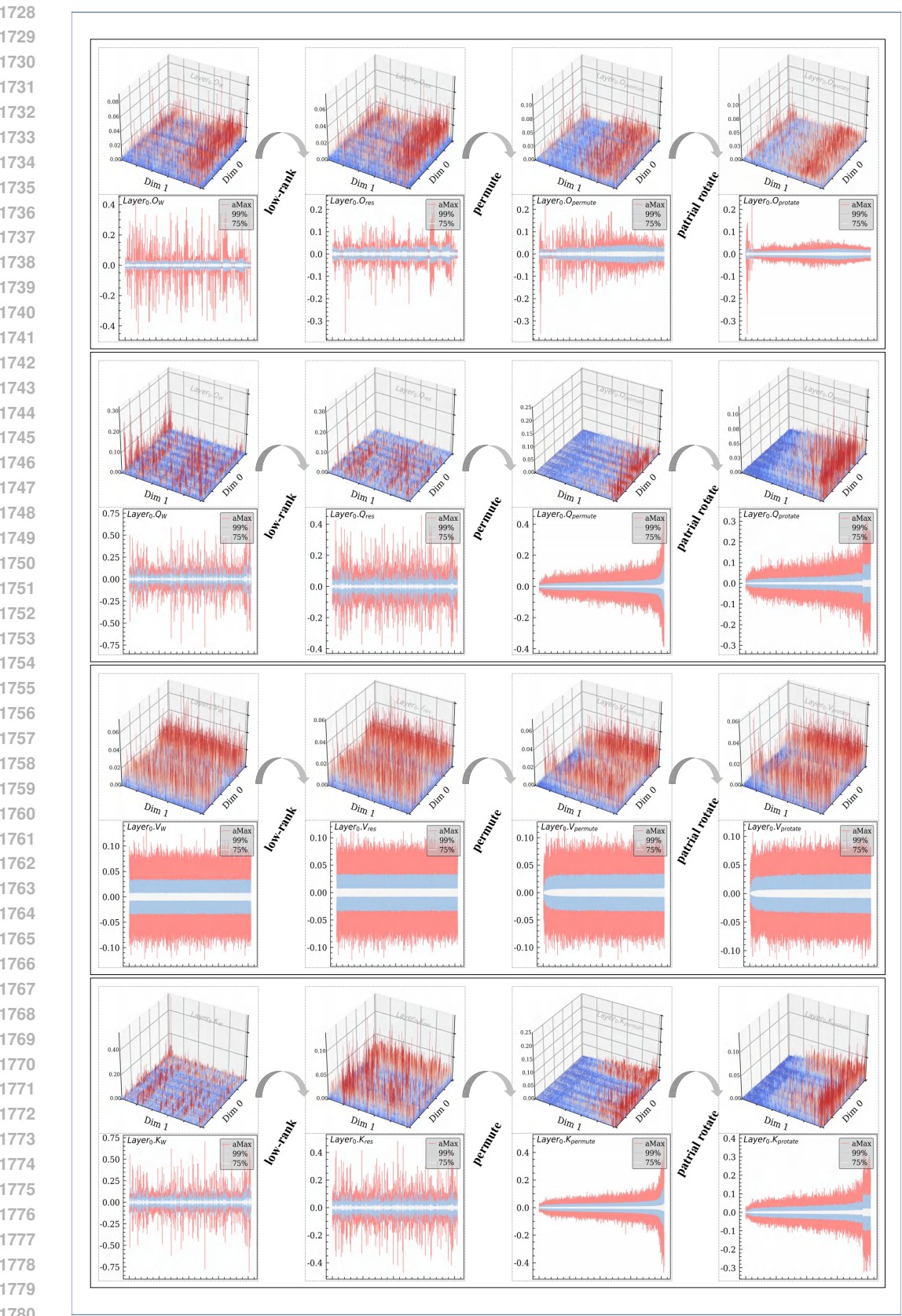

Figure 3: Visualization of layers in LLaMA2-7B

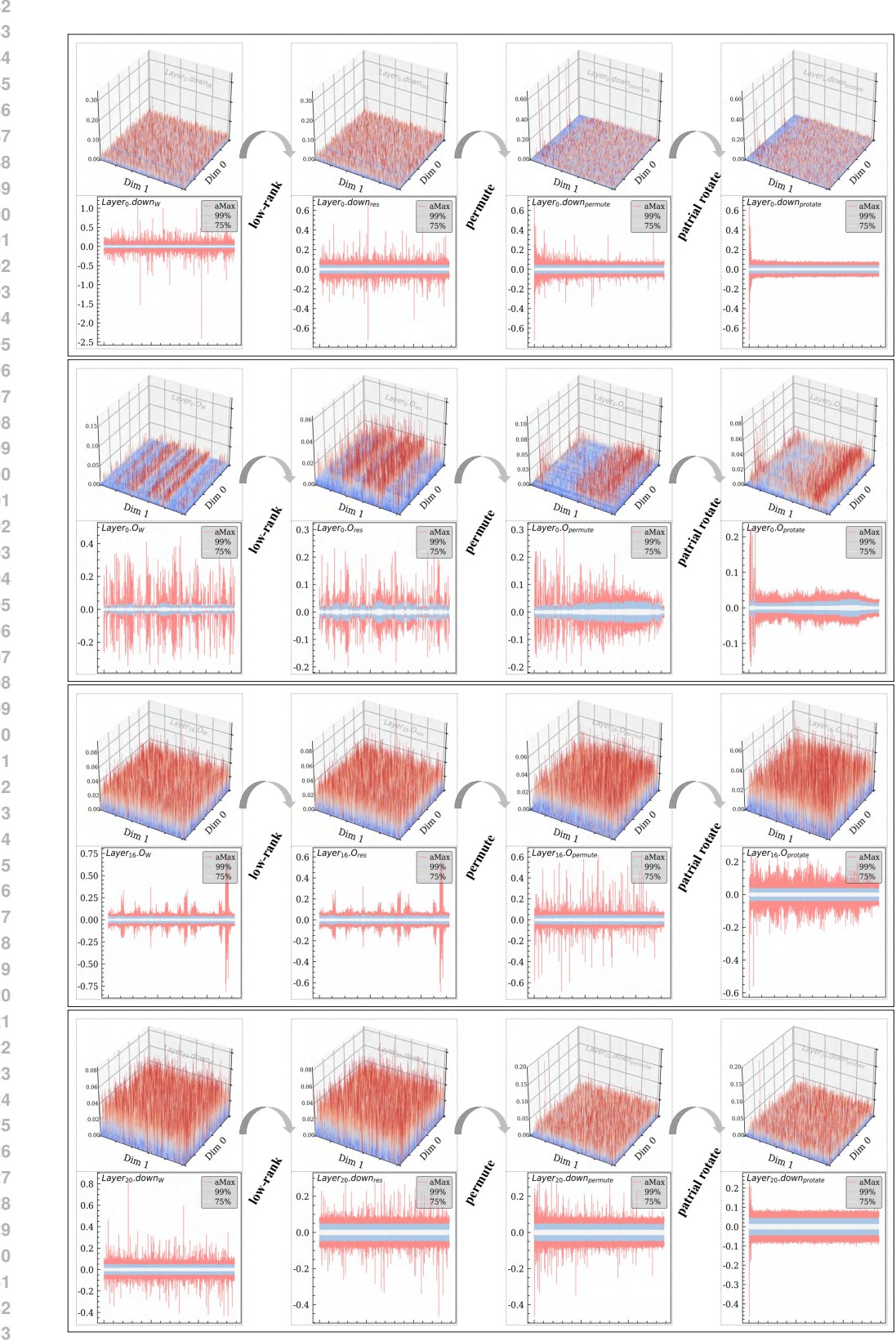

Figure 4: Visualization of layers in LLaMA2-13B

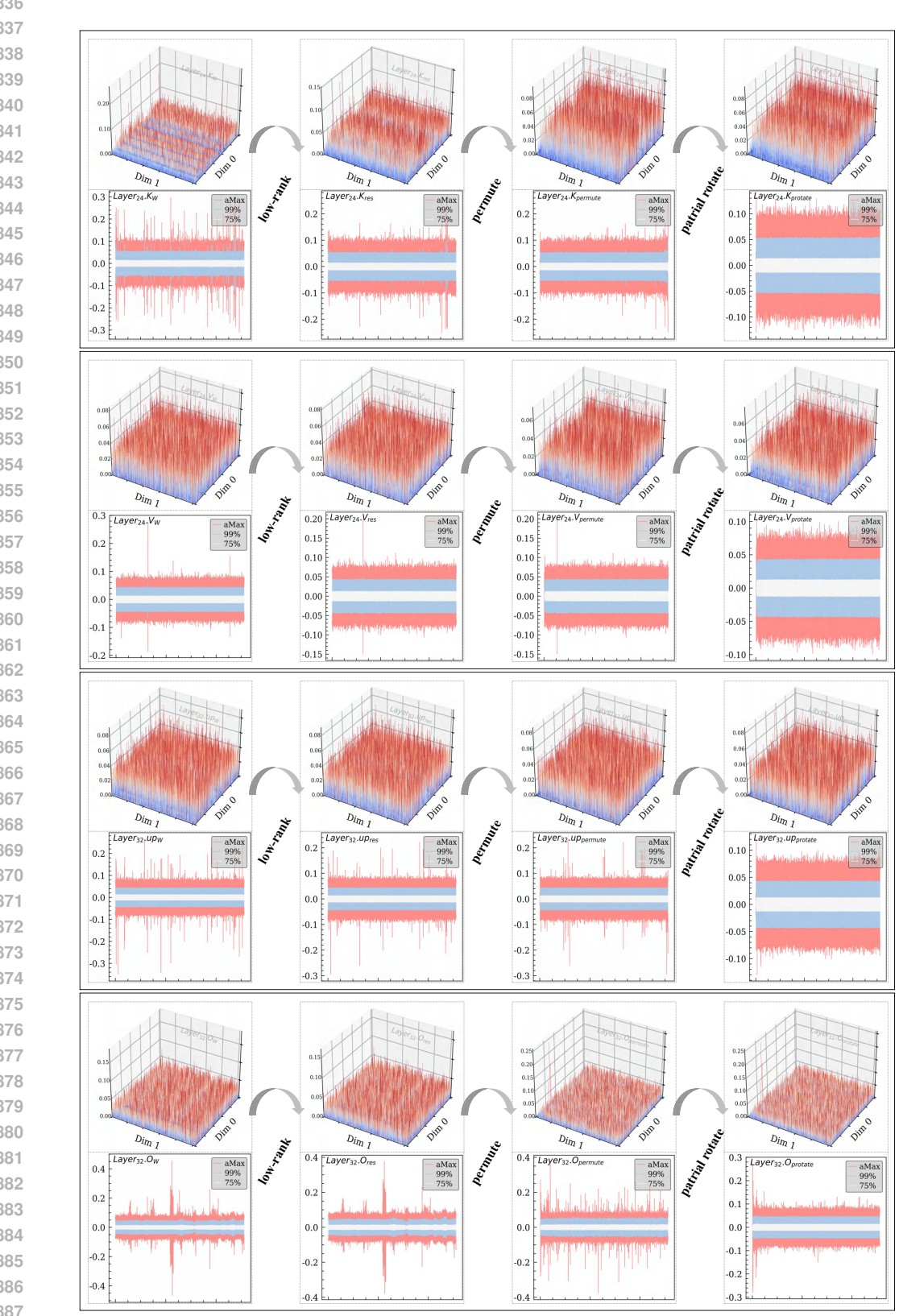

Figure 5: Visualization of layers in LLaMA2-13B

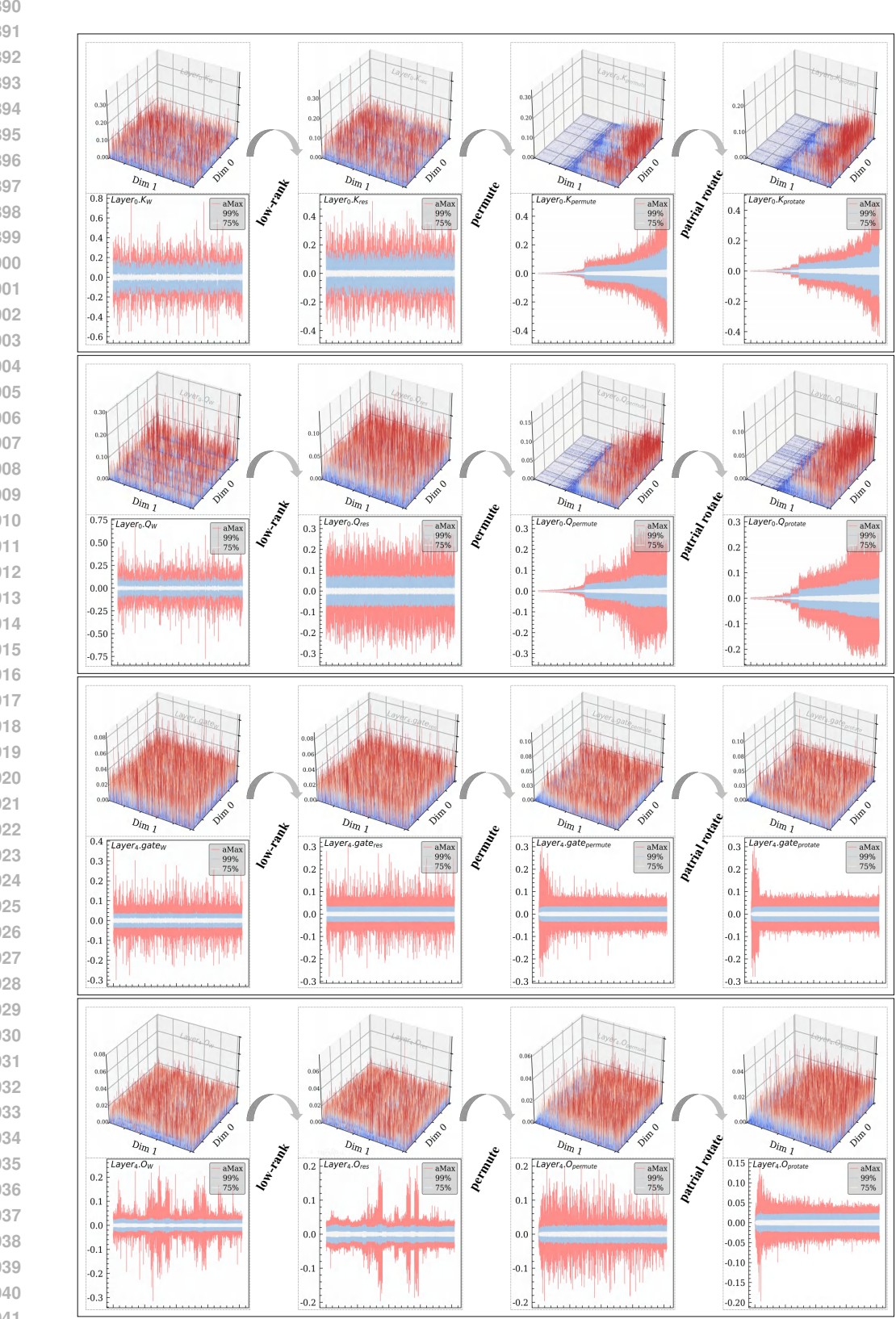

Figure 6: Visualization of layers in LLaMA3-8B

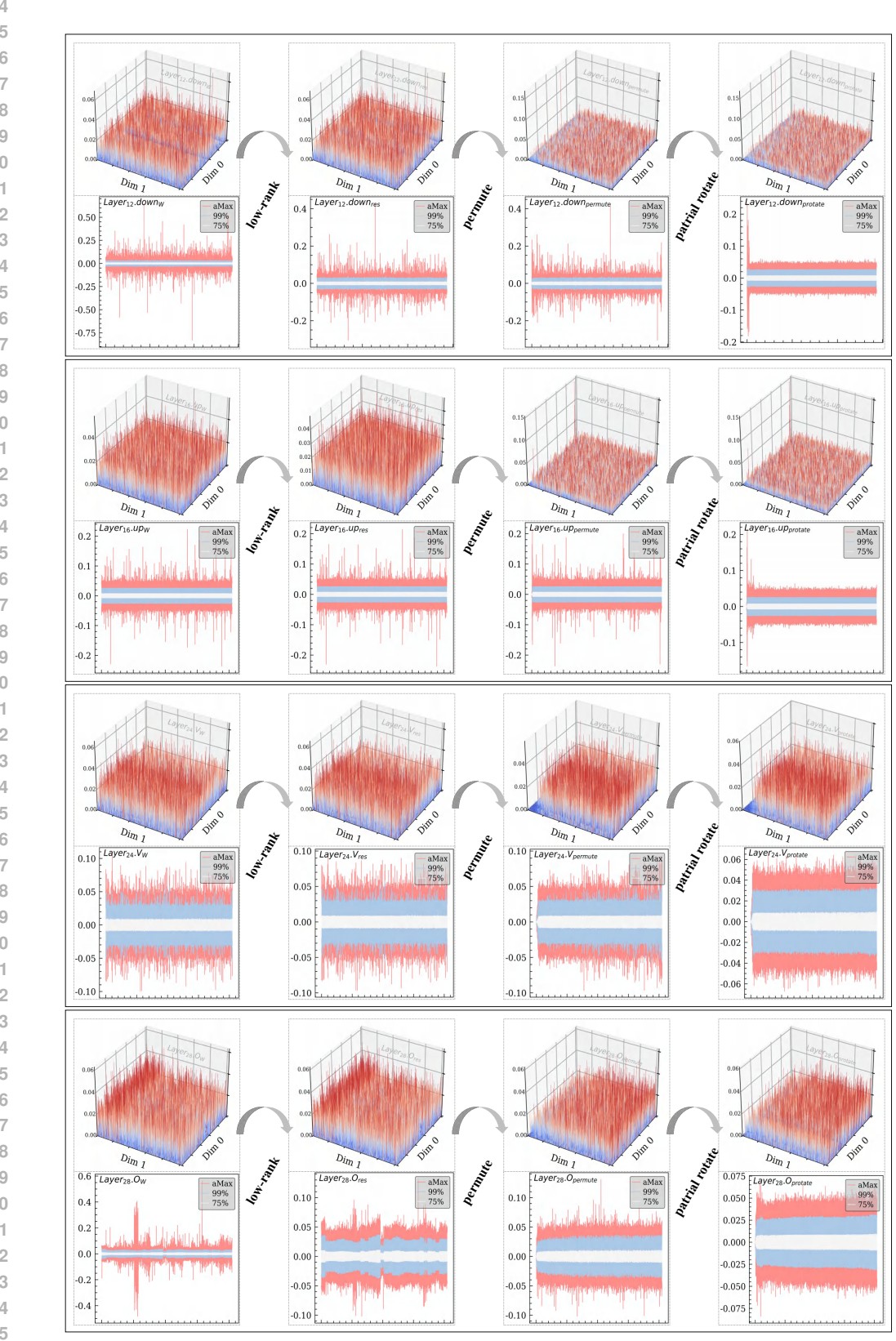

Figure 7: Visualization of layers in LLaMA3-8B

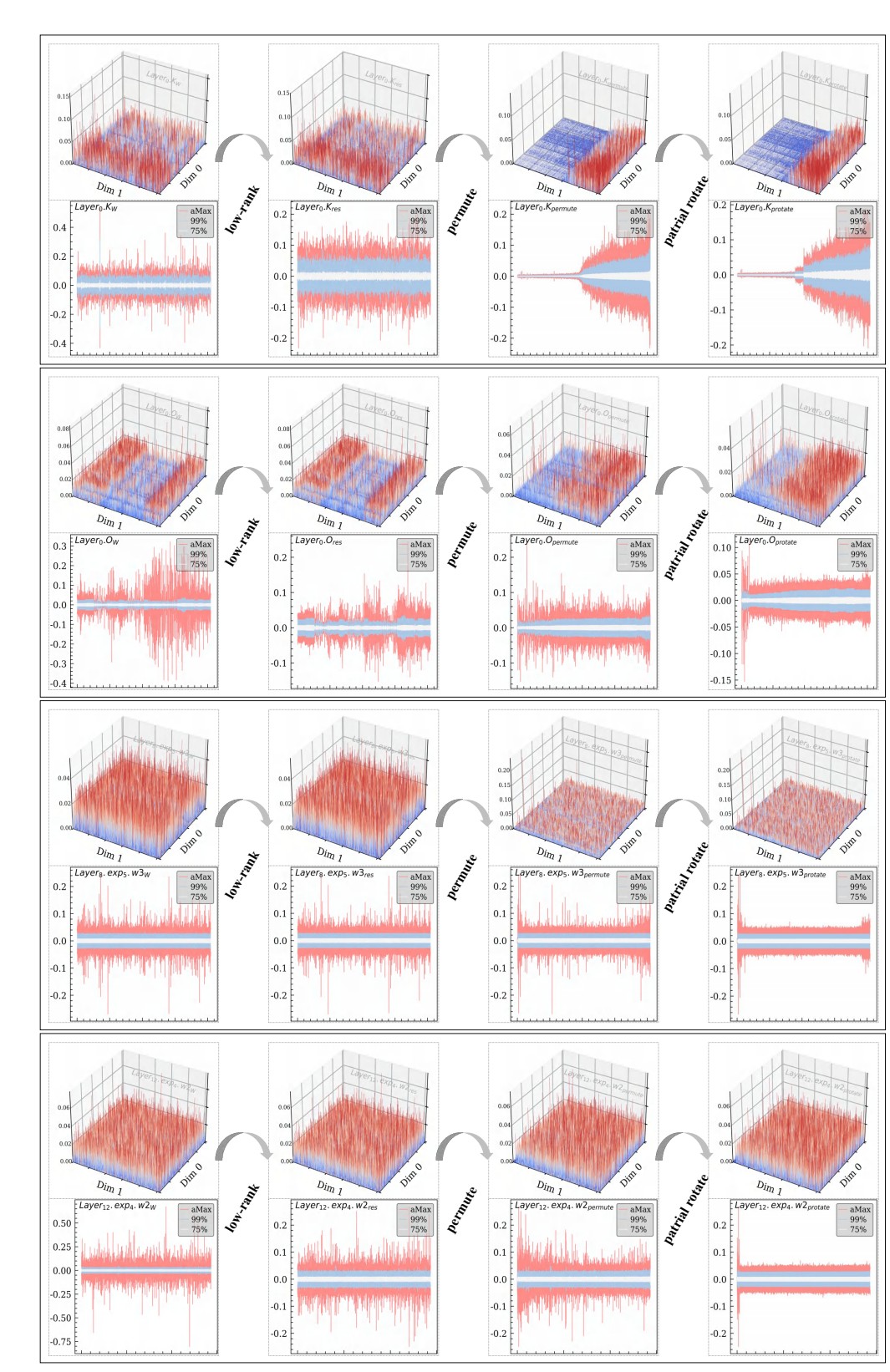

Figure 8: Visualization of layers in Mixtral-8x7B

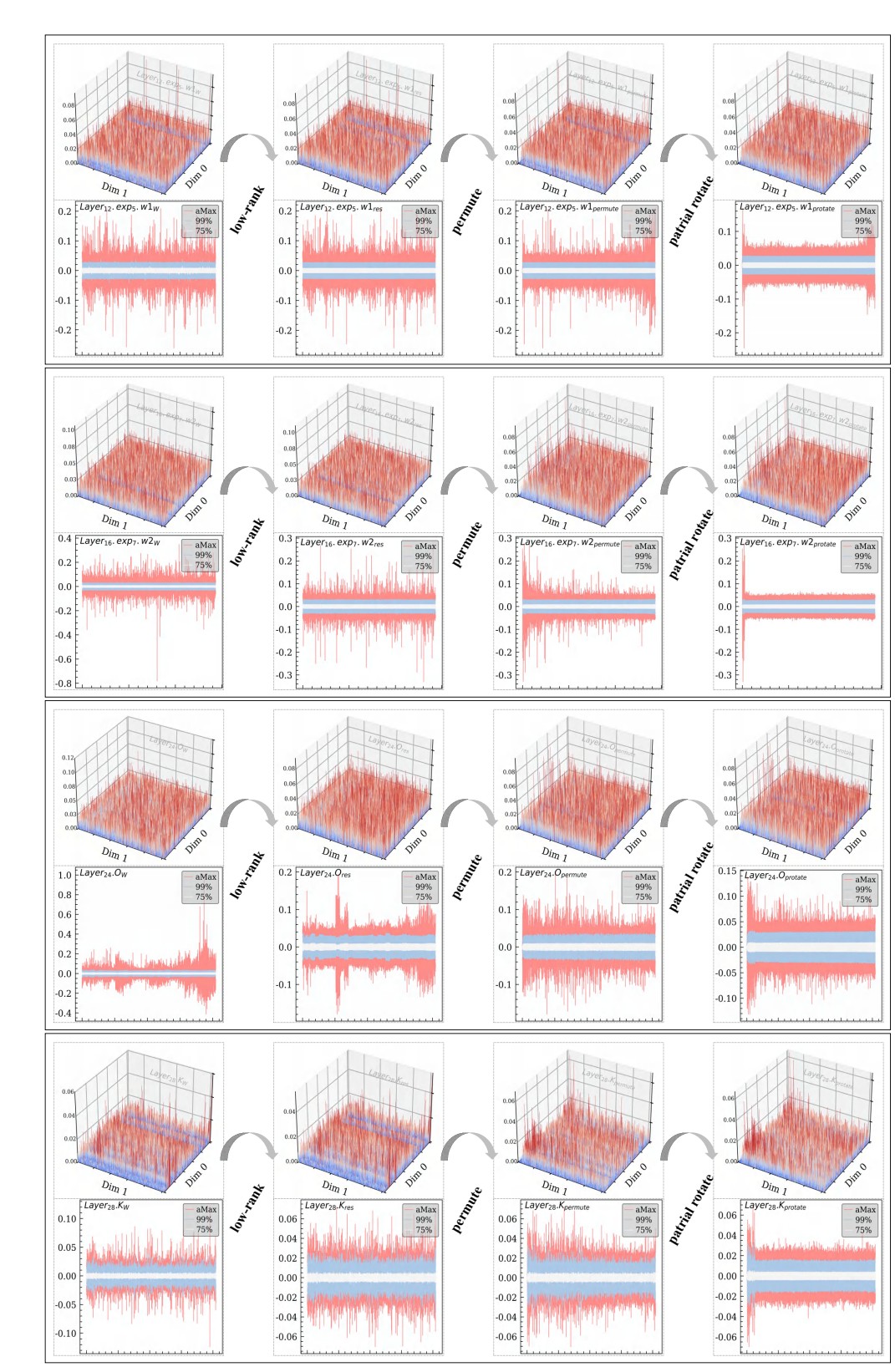

Figure 9: Visualization of layers in Mixtral-8x7B

