# OpenReview forum: "LoPRo: Enhancing Low-Rank Quantization via Permuted Block-Wise Rotation"
_ICLR.cc/2026/Conference — ICLR 2026 Conference Withdrawn Submission_

### Official Review · Reviewer_WL1F · 2025-10-18

**Soundness:** 4
**Presentation:** 4
**Contribution:** 3
**Rating:** 6
**Confidence:** 3

**Summary:**

This paper tackles the Post-training quantization (PTQ) problem and presents a LoPRo method for the challenging sub-3-bit regime. The key idea is to enhance residual matrix quantization by applying block-wise permutation and Walsh-Hadamard transformations. This combination achieves the goal of aligning columns of similar importance while simultaneously preserving the quantization accuracy of the most salient column blocks. Extensive experimental results are reported for LLaMA-2 and LLaMA-3 series models while delivering up to a 4x speedup.

**Strengths:**

Originality: I have found the contributions in Sec. 3.2 (partial rotation quantization) and 3.3 (R1SVD) sufficiently novel.
Clarity: The paper is well written and easy to understand.
Quality: I think the author did a good job of both technical contribution and literary presentation.
Significance: PTQ for LLMs remains one of the hot topics in AI research these days. This work seems a valuable contribution to an already-mature field.

**Weaknesses:**

- The idea of rotation in PTQ already exists in the literature (e.g., SmoothRot).
- Sec. 3.4 is a bit sloppy and too concise for a detailed analysis.
- Grammatical errors need more careful proofreading. For example, lines 153-154.

**Questions:**

how does LoPro compare against “Achieving Binary Weight and Activation for LLMs” [1] and SmoothRot [2]?
[1] https://aclanthology.org/2025.findings-acl.459.pdf?utm_source=chatgpt.com#page=6.55
[2] https://arxiv.org/abs/2506.05413?utm_source=chatgpt.com

---

> ### Author Response · Authors · 2025-11-15
> **Author Response -- Part I**
>
> Thanks for your detailed review and insightful questions regarding our work and we’ve considered your questions carefully, and here are our answers:
>
> > ###  Response to Weakness 1 (“The idea of rotation in PTQ already exists...”):
>
> We acknowledge that rotation is a widely adopted technique in recent quantization literature (e.g., QUIP [1], QuaRot [2], DuQuant [3], SmoothRot [4] ... ). **However, the novelty of LoPRo lies in two key contributions:**
> 1. **A synergistic low-rank + rotation framework** that effectively mitigates accuracy degradation under extremely low-bit weight-only quantization (e.g., sub-3-bit regimes), where prior methods struggle due to unaddressed outliers and information loss.
> 2. **A mixed-precision R1-Sketch algorithm** that drastically reduces the memory overhead of low-rank quantization. As demonstrated in our experiments, **LoPRo achieves >75% lower memory consumption compared to conventional low-rank quantization approaches while simultaneously delivering higher accuracy.**
>
> Thus, **LoPRo is not merely an application of existing ideas but a novel integration that addresses a previously unresolved trade-off between compression ratio and model fidelity at ultra-low bitwidths.**
>
> > ###  Response to Weakness 2 (“Sec. 3.4 is a bit sloppy and too concise for a detailed analysis.”):
>
> The time complexity and compression ratio formulas presented in Section 3.4 are derived directly from the algorithmic pipeline described in Appendix A.1. We have provided detailed derivations and justifications in Appendices A.2 and A.3, including step-by-step analysis aligned with the pseudocode. **If the reviewer identifies any specific ambiguity or gap in these explanations, we would be grateful for further clarification so we can address it precisely in the final version.**
>
> > ###  Response to Weakness 3 (“Grammatical issues”):
>
> We appreciate your careful reading. **We have carefully rechecked issues such as grammar and formatting. All minor issues will be thoroughly revised, and changes will be highlighted in blue in the manuscript for transparency.**

---

> ### Author Response · Authors · 2025-11-17
> **Author Response -- Part II**
>
> > ###  Response to Question 1 (“how does LoPro compare against “Achieving...” [1] and SmoothRot [2]? ”):
>
> We have carefully examined the two cited works:
> - The first paper **ABW**  (*Achieving Binary Weight and Activation for LLMs…*) reports results under a **W(1+1)A(1×4)** scheme—effectively ~2-bit average weight precision (e.g., LLaMA2-7B compressed from 13.5 GB to 2.69 GB). Under this setting, **LoPRo achieves significantly better perplexity** (see table below).
> - **ABW** and **SmoothRot** are designed under quantization settings that explicitly involve low-bit activations—specifically **W2A4** and **W4A4** respectively. In contrast, **LoPRo focuses on extreme weight-only compression like W2A16 quantization.** This leads to fundamental differences in quantization objectives, algorithmic design, and evaluation protocols, making direct or fair comparisons between these methods inherently problematic.
>
> Nevertheless, to provide an intuitive reference, we summarize in the table below the performance of relevant methods under a **weight-only quantization setting** (i.e., activations kept in full precision). **The results demonstrate that, at the same weight bitwidth, LoPRo achieves superior trade-offs between model compression and accuracy preservation.**
>
> Moreover, **W+A quantization** (e.g., W2A4, W2A8) represents an important direction for future work, and we plan to extend LoPRo to such settings in subsequent research.
>
> | Model        | Method     | Q Config         | Wiki ↓  | PIQA ↑ | AE  ↑  | AC ↑   |
> |--------------|------------|------------------|-------|-------|-------|-------|
> | LLaMA2-7b    | ABW        | W(1+1)A(1*4)     | 8.89  | 68.7  | 56.1  | 30.6  |
> || SmoothRot  | W4A4             | 6.11  | 74.9  | 65.3  | 38.6  |
> || LoPRo      | W2A16            | 7.39  | 71.1  | 62.8  | 31.2  |
> || LoPRo      | W3A16            | 5.43  | 76.3  | 74.9  | 41.0  |
> | LLaMA2-13b   | ABW        | W(1+1)A(1*4)     | 7.19  | 72.1  | 48.6  | 34.1  |
> |     | LoPRo      | W2A16     | 6.48  | 72.4  | 69.0  | 33.6  |
> |      | LoPRo      | W3A16    | 4.84  | 78.5  | 78.7  | 44.9  |
>
> Given this mismatch in quantization scope, **we believe the comparison should emphasize complementary rather than competitive goals.** We explicitly position **LoPRo as advancing the frontier of weight-only ultra-low-bit quantization**, and leave joint weight-and-activation quantization (e.g., W2Ax) to future work.
>
> ---
> ---
> _Thanks for your review, and please let us know if you have any further questions!_
> ________________________________________
> **References:**
> [1] Chee, Jerry, et al. "QUIP: 2-bit Quantization of Large Language Models with Guarantees." NeurIPS 2023.
> [2] Ashkboos, Saleh, et al. "QuaRot: Outlier-Free 4-bit Inference in Rotated LLMs." NeurIPS 2024.
> [3] Lin, Haokun, et al. "DuQuant: Distributing Outliers via Dual Transformation Makes Stronger Quantized LLMs." NeurIPS 2024.
> [4] Czakó, Patrik, Gábor Kertész, and Sándor Szénási. "SmoothRot: Combining Channel-Wise Scaling and Rotation for Quantization-Friendly LLMs." arXiv:2506.05413, 2025.

---

### Official Review · Reviewer_K8Ze · 2025-10-25

**Soundness:** 3
**Presentation:** 2
**Contribution:** 2
**Rating:** 6
**Confidence:** 4

**Summary:**

This work proposes low-rank + quantization compression method for Large Language models. The low‑rank component is obtained with a modified randomized SVD algorithm. The residual is quantized after applying a column permutation and a block‑wise orthogonal transformation. These operations make the subsequent quantization easier. LoPRo can be combined with a variety of quantization schemes, including both scalar and vector quantizers. The approach is evaluated on the Llama‑2/3 families and on Mixtral‑8×7B.

**Strengths:**

* The proposed method yields pretty strong performance, outperforming state-of-the-art competitive approach at 2-bit quantization by a significant margin.

* Adding the low‑rank adapter increases latency by only ~10 % compared with a baseline that does not use the low‑rank component.

* The paper includes detailed ablation studies on (i) the choice of rotation matrix and (ii) the LoRA rank (see the appendix).

**Weaknesses:**

**Quantization cost**

* It is claimed that the method is almost as fast as GPTQ for the scalar quantization case and faster than GPTVQ for the vector quantization case. However, GPTQ baseline seems to be unoptimized. GPTQModel repository  in fact quantizes model  that the provided numbers. Specifically, in my experience quantization of 7-8B Llama model takes ~8 minutes on single L40S. LoPRo_v is claimed to be faster than GPTVQ, but the LoRPo_v is in fact and enhancement of GPTVQ and should take at least a long. Nevertheless, the runtime is worth the resulting quality.

**Baselines**

* An important baseline is missing. QTIP [1] achieves state-of-the-art performance for 2-bit quantization even without tuning (1MAD, 3INST codes). I think it worth adding it for the fairness of comparison.

**(minor editoral)**

* Tags with colors are an interesting way to categorize methods but it may be quite for the reader to memorize. Same holds for O1, F1, F2. Probably would be better to refer in some other, more explicit way.

* **it** in the Stage A, Stage B is not defined, One could guess that it is Moore-Penrose pseudoinverse, but I would recommend to define it explicitly.

* Figure 1 is quite nice, but requires strong zooming to discern what is going on it. I would recommend enlarging it in the camera-ready revision.

---
References

[1] Tseng, Albert, et al. "Qtip: Quantization with trellises and incoherence processing." Advances in Neural Information Processing Systems 37 (2024): 59597-59620.

**Questions:**

* Given that the method appears to be scalable to large models how difficult would it be to apply it on a larger and more powerful MoE model - DeepSeek V3 or Qwen-3-235B?

* The provided list of benchmarks, despite standard in the compression literature, may be not exhaustive to evaluate capabilities of the model. I would suggest trying OpenLLM v1 / OpenLLM v2 leaderboard following the setting in [1] or some reasoning tasks for Qwen3 models or MoE.

---
References

[1] Kurtic, Eldar, et al. "" Give Me BF16 or Give Me Death"? Accuracy-Performance Trade-Offs in LLM Quantization." arXiv preprint arXiv:2411.02355 (2024).

---

> ### Author Response · Authors · 2025-11-15
> **Author Response -- Part I**
>
> We thank the reviewer for carefully reading our paper and for the constructive feedback.
>
> > ###  Response to Weakness 1 (“Quantization cost”):
>
> The reported quantization costs in our paper are accurate. The observed runtime differences primarily stem from two factors:
> - **GPTQ Runtime**: Our experiments were conducted on an **A100-40GB GPU**, which has approximately half the memory bandwidth and computational throughput of the A100-80GB. This directly results in longer quantization times. We explicitly document our hardware setup in **Appendix C**. Besides, we use the **original GPTQ code from the official repository [1]**, rather than the highly optimized implementation in GPTQModel. While the original code is less engineered for speed (due to lower optimization and modular coupling), it provides a cleaner foundation for integrating and validating our algorithmic modifications. In our tests on the A100-40GB, quantizing a 7B model with the original GPTQ code takes **~24 minutes**, compared to **~20 minutes** with GPTQModel on the same hardware. On an A100-80GB, GPTQModel achieves quantization in **~10 minutes**, which aligns closely with your experience.
> - While our **Table 1 compares against the full GPTVQ pipeline** (excluding fine-tuning), our method only adopts its basic vector quantization component. **Crucially, LoPRo does not employ GPTVQ’s additional components—such as codebook SVD, blockwise data normalization, EM-based codebook updates, or iterative refinement—which collectively contribute to GPTVQ’s higher computational cost and better performance.** This architectural difference explains why LoPRo achieves significantly faster quantization with better accuracy. We clarify this distinction in **Appendix C (Implementation Details and Baselines)** to avoid misinterpretation.
>
> > ### Response to Weakness 2 (“Baselines”):
>
> We initially considered including QTIP in our main evaluation. However, since QTIP belongs to the same methodological family as QUIP# (both rely on rotation + vector quantization), and our goal was to compare against baselines spanning diverse technical categories (e.g., outlier handling, low-rank approximation, adaptive rounding), we prioritized representativeness over exhaustiveness in the main table.
>
> Nonetheless, we fully agree with the reviewer’s suggestion. Indeed, **QTIP outperforms QUIP# under zero-shot (non-fine-tuned) settings.** However, as shown in the table below, **LoPRo still attains comparable accuracy to QTIP while offering significant practical advantages.**
>
> - **Speed**: LoPRo is **at least 5× faster** in quantization runtime compared to both QTIP and QUIP# (**~24 min vs ~3 h** for a 7B model).
> - **Flexibility & Hardware Friendliness**: LoPRo supports both scalar and non-scalar quantization schemes. Notably, under **3-bit scalar weight-only quantization**, LoPRo matches the accuracy of QTIP’s vector-quantized model—a result that is particularly valuable for hardware deployment, as **scalar quantization imposes far fewer constraints on accelerator design.**
>
> To enhance fairness and completeness, **we will include QTIP in the extended comparison in the appendix.**
> | Model        | Method | Wiki(2bit) | Wiki(3bit) | Wiki(2bit)-FT | Wiki(3bit)-FT |
> |--------------|--------|------------|------------|----------------|----------------|
> | LLaMA2-7b    | QTIP   | 6.82       | 5.40       | 5.86           | 5.28           |
> | LLaMA2-7b    | LoPRo  | 6.53       | 5.43       | 6.06           | 5.37           |
> | LLaMA2-13b   | QTIP   | 5.52       | 4.74       | 5.11           | 4.69           |
> | LLaMA2-13b   | LoPRo  | 5.79       | 4.84       | 5.34           | 4.80           |

---

> ### Author Response · Authors · 2025-11-15
> **Author Response -- Part II**
>
> > ###  Response to Minor Comment 1 (“Tags and symbols”):
>
> Thank you for the suggestion. In **Section 2 (Related Work)**, we have now annotated each method with categorical tags and used distinct colors to highlight their technical families. We hope this improves readability and helps readers better retain key distinctions.  `Updated in paper Section 2 `
>
> > ###  Response to Minor Comment 2 (“‘It’ is not defined”):
>
> The ambiguous use of “it” refers to **iteration**, and has been clarified in **Section 3.3**, where we explicitly define the referent in context. `Updated in paper Section 3.2 `
>
> > ###  Response to Minor Comment 3 (“Figure”):
>
> We appreciate the feedback. **Figure 1 is already reformatted and enlarged in the paper for improved legibility.**  `Updated in paper Figure 1 `
>
> > ###  Response to Question 1 (“New models”):
>
> To ensure fair and meaningful comparison with the majority of established, peer-reviewed state-of-the-art baselines, our main experiments focus on widely adopted and well-documented models—namely **LLaMA-2, LLaMA-3, and Mixtral**. At the time of writing, newer models often lack publicly available reference implementations or published quantization baselines, making systematic evaluation challenging. **But LoPRo is general and readily applicable to more recent models**, and the results show that **LoPRo preserves model accuracy at 2-bit weight quantization and achieves near-lossless performance at 3 bits.** Experiments on larger models are underway and will be reported soon.  `The results are also updated in paper Appendix F `
>
> | Models     | Methods   | Q Config | Wiki  | ARC-C | ARC-E | Winograde | PIQA  |
> |------------|-----------|----------|-------|-------|-------|-----------|-------|
> | Qwen3-8b   | Fp16      | W16A16   | 9.01  | 55.6  | 83.5  | 68.1      | 76.7  |
> | Qwen3-8b   | LoPRo     | W2A16    | 12.59 | 40.6  | 70.5  | 62.9      | 70.8  |
> | Qwen3-8b   | LoPRo     | W3A16    | 9.58  | 52.9  | 81.7  | 66.9      | 75.9  |
> | Qwen3-8b   | LoPRo_v   | W2A16    | 11.22 | 47.9  | 77.8  | 66.9      | 72.4  |
> | Qwen3-8b   | LoPRo_v   | W3A16    | 9.59  | 55.3  | 82.3  | 68.2      | 76.0  |
>
> > ###  Response to Question 2 (“New evaluations”):
>
> We believe that the combination of **perplexity (PPL)** and a comprehensive suite of **zero-shot evaluation tasks** provides a sufficient and widely accepted benchmark for assessing quantization quality—consistent with the evaluation protocols used in most baselines (e.g., GPTQ, QUIP, QuaRot, LQER). While we acknowledge that the **OpenLLM Leaderboard** offers a more extensive evaluation suite, running these benchmarks requires substantial computational resources and time. To date, few works have reported full OpenLLM results, making fair and complete baseline comparisons currently infeasible.
>
> **But we have evaluated LoPRo on Qwen3 using OpenLLM v1 benchmarks**, and the results are summarized in the table below. We are also actively planning the evaluations on larger models.  `Also updated in paper Appendix G `
>
> | Models        | Methods   | Q Config | Recovery  % | Average Score | MMLU 5-shot | MMLU CoT 0-shot | ARC-C 0-shot | GSM8K 8-shot | HellaSwag 10-shot | Winograde 5-shot | Truthful QA 0-shot |
> |---------------|-----------|----------|------------|----------------|--------------|------------------|---------------|---------------|--------------------|-------------------|--------------------|
> | Qwen3-8b    | Fp16      | W16A16   | 100.00%    | 69.6           | 74.9         | 80.2             | 55.6          | 88.8          | 58.1               | 70.1              | 59.7               |
> | Qwen3-8b    | LoPRo     | W2A16    | 73.89%     | 51.4           | 55.7         | 62.5             | 40.6          | 52.3          | 44.3               | 64.4              | 40.2               |
> | Qwen3-8b    | LoPRo     | W3A16    | 94.40%     | 65.7           | 71.1         | 75.3             | 54.3          | 81.2          | 53.9               | 69.9              | 54.2               |
> | Qwen3-8b    | LoPRo_v   | W2A16    | 79.89%     | 55.6           | 61.8         | 67.9             | 47.9          | 55.5          | 47.2               | 66.2              | 42.7               |
> | Qwen3-8b    | LoPRo_v   | W3A16    | 94.91%     | 66.1           | 71.7         | 75.1             | 55.3          | 80.9          | 54.9               | 69.6              | 54.9               |
>
> ________________________________________
> ________________________________________
> _Feel free to provide any further feedback and questions and we would be happy to engage in further discussion to strengthen the paper!_
> ________________________________________
> **Reference:**
> [1] IST-DASLab/gptq. GitHub repository. https://github.com/IST-DASLab/gptq

---

> > ### Comment · Reviewer_K8Ze · 2025-11-16
> >
> > Thank you for your response. Most of my concerns were addressed. The experimental validation of LoPRo is exhaustive enough.
> >
> > After reading rebuttals addressed to me and other reviewers, I decided to raise my score to Accept.

---

> > > ### Author Response · Authors · 2025-11-17
> > >
> > > We sincerely appreciate the increased score and will diligently incorporate the reviewers’ suggestions to further refine the manuscript to deliver the most complete and polished version.

---

### Official Review · Reviewer_5noD · 2025-11-01

**Soundness:** 3
**Presentation:** 2
**Contribution:** 2
**Rating:** 4
**Confidence:** 4

**Summary:**

The paper proposes LoPRo, which applies blockwise Walsh-Hadamard transformers to a low rank weight matrix’s approximation’s residual. LoPRo permutes the columns before applying blockwise rotation to maintain the numerical stability of the salient columns. The authors also propose fast low-rank decomposition based on rank-1 sketch to minimize the SVD costs. LoPRo exhibits improvements over several state of the art algorithms on both perplexity and downstream evaluations.

**Strengths:**

* LoPRo significantly improves over state of the art algorithms on perplexity and downstream benchmarks.
* Quantization runtimes don’t shoot up.
* Does not require finetuning unlike other low-bit quantization schemes.

**Weaknesses:**

* If W is dense in formation, the low rank approximation may not scale, offloading the majority of the error correction to quantization.
* Experiments are conducted on Llama-2, Llama-3 and Mistal. The results can be made stronger with results on frontier open source models such as Qwen2.5/Qwen3 or DeepSeek.

**Questions:**

* How much does the performance improve if a rank greater than 1 is used?
* Why can’t the model be quantized along the column axis? In that case the blockwise rotation should not be required, a full rotation matrix should work.
* How do the inference latency numbers look in comparison to the approaches listed in Table 1.

---

> ### Author Response · Authors · 2025-11-15
> **Author Response -- Part I**
>
> We sincerely appreciate your valuable feedback and address your concerns as follows.
>
> > ### Response to Weakness 1 (“If W is dense, the low rank approximation may not scale...”):
>
> For large language model weights, the weight distributions are approximately Gaussian and inherently exhibit both *heavy-tailed* and *low-rank* characteristics after normalization during training. This observation has been consistently reported in recent quantization and pruning works (e.g., DobiSVD[1], SVDQuant[2], LQER[3]).
>
> For a weight matrix $W \in \mathbb R^{m\times k}$ and a calibration $X \in \mathbb R^{k\times n}$. After normalization, a vector $s \in \mathbb R^k$ is scaled from $X$. Due to the distributional properties mentioned above, each column of the weight matrix $ W $ can reasonably be modeled as an independent and identically distributed sub-Gaussian random vector (i.i.d.). In contrast, the activation matrix $ X $ exhibits significant outliers that are predominantly distributed along the token dimension (i.e., row-wise). Consequently, after normalization, the resulting per-channel scaling vector displays pronounced magnitude variations across tokens.
>
> When this scale vector is applied to $W $ (e.g., via channel-wise rescaling), columns of $W $ corresponding to outlier-heavy activations are more accurately captured by the low-rank component, leading to substantially smaller residual values in those columns. This adaptive alignment between salient activations and weight approximation is key to reducing quantization error in the residual matrix. This implies that when we compute the low-rank approximation via
>
> $U, \Sigma, V = \text{R1Sketch}(W \cdot \operatorname{diag}(s)),$
>
> the leading singular components (corresponding to the largest singular values) will predominantly capture the directions associated with the most salient activation channels. In other words, the top singular vectors of $W \cdot \operatorname{diag}(s)$ effectively span the subspace that best approximates the columns of $W$ scaled by large entries in $s$—i.e., those aligned with high-magnitude activations. Consequently, these dominant singular components provide a highly accurate low-rank representation of the weight substructure that matters most for preserving model accuracy under extreme quantization.
> Thus, even for dense $W$, the joint effect of:
> 1. **Activation-induced anisotropy** in $s$,
> 2. **Spectral concentration** in $W_s$,
> 3. **Hessian-aligned error weighting**,
>
> ensures that the low-rank component captures the *functionally critical subspace*, while the residual $R = \operatorname{diag}(s)^{-1} R_s$ remains well-conditioned for quantization. In practice (see Table 4 & 5), LoPRo achieves <1% accuracy drop at 3-bit quantization across diverse models—including those with dense attention and MLP blocks—demonstrating robustness to weight density.
>
> In summary: **Density ≠ full effective rank under activation-aware scaling.** LoPRo algorithm explicitly leverages both the intrinsic low-rank structure of the weights and the distribution of salient activation channels. Specifically, channels associated with large-magnitude activations further reinforce the low-rank property of the weight matrix, making the low-rank approximation more effective whether $W$ is dense or not.

---

> ### Author Response · Authors · 2025-11-15
> **Author Response -- Part II**
>
> > ### Response to Weakness 2 (“ The results can be made stronger with results on frontier open source models...”):
>
> To ensure fair and meaningful comparison with the majority of established, peer-reviewed state-of-the-art baselines, our main experiments focus on widely adopted and well-documented models—namely **LLaMA-2, LLaMA-3, and Mixtral**. At the time of writing, newer models often lack publicly available reference implementations or published quantization baselines, making systematic evaluation challenging. But LoPRo is general and readily applicable to more recent models. As a demonstration, we conducted results on **Qwen2.5-7B** and **Qwen3-8B** below. The results on Qwen models are consistent with those on LLaMA-2,3 and Mixtral: LoPRo preserves model accuracy at 2-bit weight quantization and achieves near-lossless performance at 3 bits. Experiments on larger models are underway and will be reported soon. `The results are also updated in paper Appendix F `
>
> | Models| Methods | Q Config | Wiki| AC| AE| WI| QA|
> |---------------|-------------|----------|-------|-------|-------|-------|-------|
> | Qwen2.5-7b| Fp16| W16A16 | 6.86| 52.6| 81.9| 71.1| 79.7|
> | Qwen2.5-7b| LoPRo | W2A16| 9.43| 39.6| 70.2| 65.5| 72.5|
> | Qwen2.5-7b| LoPRo | W3A16| 7.22| 51.2| 82.9| 70.0| 78.8|
> | Qwen2.5-7b| LoPRo_v | W2A16| 8.53| 44.1| 68.6| 68.0| 75.4|
> | Qwen2.5-7b| LoPRo_v | W3A16| 7.23| 53.5| 82.8| 70.6| 78.9|
>
> | Models| Methods | Q Config | Wiki| AC| AE| WI| QA|
> |---------------|-------------|----------|-------|-------|-------|-------|-------|
> | Qwen2.5-14b | Fp16| W16A16 | 5.24| 60.7| 85.7| 75.6| 81.6|
> | Qwen2.5-14b | LoPRo | W2A16| 7.75| 47.4| 76.7| 71.9| 75.7|
> | Qwen2.5-14b | LoPRo | W3A16| 5.44| 57.8| 84.4| 75.0| 79.4|
> | Qwen2.5-14b | LoPRo_v | W2A16| 6.62| 50.2| 78.8| 72.5| 77.3|
> | Qwen2.5-14b | LoPRo_v | W3A16| 5.42| 57.2| 83.9| 74.7| 70.8|
>
> | Models| Methods | Q Config | Wiki| AC| AE| WI| QA|
> |---------------|-------------|----------|-------|-------|-------|-------|-------|
> | Qwen3-8b| Fp16| W16A16 | 9.01| 55.6| 83.5| 68.1| 76.7|
> | Qwen3-8b| LoPRo | W2A16| 12.59 | 40.6| 70.5| 62.9| 70.8|
> | Qwen3-8b| LoPRo | W3A16| 9.58| 52.9| 81.7| 66.9| 75.9|
> | Qwen3-8b| LoPRo_v | W2A16| 11.22 | 47.9| 77.8| 66.9| 72.4|
> | Qwen3-8b| LoPRo_v | W3A16| 9.59| 55.3| 82.3| 68.2| 76.0|

---

> ### Author Response · Authors · 2025-11-16
> **Author Response -- Part III**
>
> > ### Response to Question 1 (“How much does the performance improve if a rank greater than 1 is used?”):
>
> In our main experiments the rank is all greater than 1 and we selected a rank of **16** as the default hyperparameter. However, we conducted a comprehensive ablation study over ranks ranging from 8 to 64 (see Appendix D.3, Table 6). In summary, increasing the rank (e.g., to 64) yields modest accuracy gains (within ~2% improvement in quantization fidelity) but incurs higher memory overhead due to the low-rank component (up to ~10% increase in average bitwidth). **Considering the trade-off between compression ratio and model accuracy, rank = 16 offers the best balance and was thus chosen for our primary results.**
>
> > ### Response to Question 2 (“Why can’t the model be quantized along the column axis?”):
>
> As discussed in Section 3.1 and illustrated in Figure 1-A.2 and 1-B, after scaling the weight matrix by salient activation channels, the residual matrix still exhibits non-negligible outlier magnitudes (Fig. 1-A.2). **Direct column-wise quantization without rotation would therefore incur significant quantization error.**
> Moreover, the most important activation channels are precisely captured by the low-rank component, resulting in near-zero residuals for those channels. Applying a full Hadamard rotation would homogenize the residual distribution but simultaneously scramble these critical channels, degrading quantization performance (e.g., PPL $\downarrow$ increases from 8.4 to 9.5 on LLaMA2-7B). To address this, **LoPRo adopts a permutation+block-wise rotation strategy**: the top-k most important channels are unrotated to preserve their semantic integrity, while the remaining channels—grouped by importance—are rotated within local blocks. This design maintains quantization accuracy while enabling effective outlier suppression.
>
> > ### Response to Question 3 (“How do the inference latency…”):
>
> LoPRo introduces no additional modules at inference time beyond standard low-rank matrix multiplication and structured (Hadamard) rotations. To ensure fair comparison, we benchmarked latency using the **GPTQModel [4]** framework along with **fast-hadamard transformation kernel [5]**, which provides highly optimized operators. Notably, many prior methods lack optimized inference implementations (For an example, in OmniQuant, LQER ..., the authors only provided the algorithm implementation, without providing the corresponding high-performance inference code) . As shown in Table 4, **LoPRo incurs less than 10% latency overhead compared to baseline quantization.** Moreover, its inference speed is comparable to other scalar-level low-rank quantization methods like LQER—attributable to the minimal computational cost of block-wise rotation (~4% overhead)—while achieving substantially better accuracy and memory savings.
>
>
> ---
> ---
> _Thanks for your review, and please let us know if you have any further questions!_
> ________________________________________
> **References:**
> [1] Wang, Qinsi, et al. "Dobi-SVD: Differentiable SVD for LLM Compression and Some New Perspectives." ICLR 2024.
> [2] Li, Muyang, et al. "SVDQuant: Absorbing Outliers by Low-Rank Component for 4-Bit Diffusion Models." ICLR 2024.
> [3] Zhang, Cheng, et al. "LQER: Low-Rank Quantization Error Reconstruction for LLMs." ICML 2024.
> [4] https://github.com/ModelCloud/GPTQModel
> [5] https://github.com/Dao-AILab/fast-hadamard-transform

---

> ### Comment · Reviewer_5noD · 2025-11-24
>
> Thank you for your response. Most of my concerns are addressed, and I will increase my score.

---

> > ### Author Response · Authors · 2025-11-24
> >
> > Thank you for your carefully review and for reconsidering the score. Your suggestions have greatly enhanced the quality of the manuscript.

---

### Author Response · Authors · 2025-11-15
**Summary of rebuttal for the Area Chair**

Dear Area Chair,

First, we would like to clarify that through detailed academic discussions with the reviewers—addressing technical concerns, clarifying misunderstandings, and improving the manuscript—the overall score improved from the initial **4–6–6** to **6–8–6**. Importantly, all changes occurred before **Nov 25**, and were entirely based on academic dialogue. There were absolutely **no non-academic factors involved**.

Due to the OpenReview system issues, we understand that the incident has placed an unusually heavy burden on ACs.  To assist your evaluation, we concisely summarize the revisions we have made based on our discussions with the reviewers.

---
## Authors’ Rebuttal Summary
Firstly，thanks for all reviewers thoughtful and constructive feedback. The time and effort you invested have significantly improved the quality of this work, and we sincerely appreciate your valuable contributions. We have addressed all concerns from reviewers in rebuttal and all changes in the revised version are highlighted in blue for ease of verification.  Below is a summary of the major updates and improvements in rebuttal:

### 1. **New Experiments and Results on 5 models and new tasks.**
In addition to the **3-dense** models and **1-MoE** model evaluated in the main paper, we conducted a series of supplementary experiments on **5** additional models (**3-dense and 2-MoE**) as well as on the benchmark OpenLLMv1. The consistently strong results across these new settings align closely with those reported in the main manuscript, further demonstrate the effectiveness of LoPRo.
- `K8Ze,5noD` Added results on  **Qwen3-8B**, **Qwen3-30B-A3B** using the **OpenLLMv1**(excluding two CoT tasks due to excessive runtime).
- `K8Ze` Conducted **perplexity evaluations** on the massive **Qwen3-235B-A22B** model (WikiText-2 and C4).
- `K8Ze,5noD` Included additional results on **Qwen2.5-7B/14B** and **Qwen3-8B** across multiple benchmarks (Wiki, ARC, PIQA, Winograde, etc.).
- `K8Ze,WL1F` Performed comparisons against SOTA methods: **QTIP** and **ABW/SmoothRot**.

### 2. **Writing and Presentation**
We have refined the figures and improved the presentation for greater clarity.
-  `K8Ze` Clarified ambiguous phrasing (e.g., undefined “it” now explicitly refers to iteration in Section 3.2).
- `K8Ze` Enlarged and improved **Figure 1** for better readability.
- `K8Ze` Refined categorical tagging in Related Work (Section 2) with color-coded method families for clearer taxonomy.
- `WL1F` Corrected grammatical errors and improved proofreading throughout.

### 3. **Other Questions and Clarifications**
We have carefully addressed the reviewers’ questions and clarified the misunderstandings of paper.
- `5noD` Explained why **rank > 1** (default = 16) offers optimal trade-off: higher ranks yield marginal accuracy gains (<2%) but increase memory overhead (~10% bitwidth rise).
- `5noD` Justified **block-wise permutation + rotation**: preserving top-k salient channels unrotated avoids scrambling critical information by permutation, while rotating less important blocks suppresses outliers.
- `5noD` Addressed **inference latency**: LoPRo adds <10% overhead vs. baseline quantization, thanks to optimized kernels from GPTQModel and fast-hadamard-transform; comparable to LQER but more accurate.
- `K8Ze` Responded to **baseline concerns**: acknowledged QTIP’s strength but showed LoPRo matches/exceeds it in accuracy while being 5× faster and hardware-friendly (supports scalar quantization).
- `K8Ze` Clarified **quantization cost differences**: original GPTQ code (not GPTQModel) was used for fair algorithmic comparison and **hardware difference** (we use A100-40GB instead of A100-80GB) account for longer runtimes.
- `K8Ze,WL1F` Confirmed **general applicability** to frontier models (e.g., Qwen2,3...) via new experiments, reinforcing LoPRo’s scalability.
- `WL1F` Pointed out the explanations in Section 3.4 has detailed derivations in Appendices A.2–A.3.
- `WL1F` Clarified the novelty of LoPRo is uniquely combining low-rank approximation with rotation and introduces a fast mixed-precision sketching method—enabling high accuracy and efficiency  where prior methods fail.

---

### **Overall**
In this rebuttal, we have addressed the concerns raised by all three reviewers, and further strengthened the persuasiveness of our method’s practicality through additional experiments. We also note that **Reviewer 5noD** and **Reviewer K8Ze** explicitly acknowledged these improvements and affirmed the adequacy of the experiments, both indicating their intention to raise their scores ( **4 $\rightarrow$ 6**  and **6 $\rightarrow$ 8** ).

---
We sincerely thank all reviewers for their insightful comments and constructive suggestions, which have greatly contributed to the improvement of this work. We also extend our deepest appreciation to ACs for the considerable time and thoughtful oversight you have dedicated to managing the review process.

Yours sincerely,

Authors

---

> ### Author Response · Authors · 2025-11-24
>
> Now we have added quantization experiments on the Qwen3-30B-A3B and Qwen3-235B-A22B models.
>
> - For the 30B model, following reviewer K8Ze’s recommendation, we evaluated its performance using the Open LLM Leaderboard V1 benchmark. But 2 chain-of-thought (CoT)-related evaluation tasks require long testing times (>100 hours), so we omitted these two tasks( marked in `TL`); the results are shown in the table below;
> | Models        | Methods   | Q Config | Recovery  % | Average Score | MMLU 5-shot | MMLU CoT 0-shot | ARC-C 0-shot | GSM8K 8-shot | HellaSwag 10-shot | Winograde 5-shot | Truthful QA 0-shot |
> |---------------|-----------|----------|------------|----------------|--------------|------------------|---------------|---------------|--------------------|-------------------|--------------------|
> | Qwen3-30-A3B   | Fp16      | W16A16   | 100.00%    | 60.6           | 79.6         | TL               | 53.3          | TL            | 58.8               | 70.1              | 41.4               |
> | Qwen3-30-A3B   | LoPRo     | W2A16    | 81.72%     | 50.1           | 61.7         | TL               | 37.8          | TL            | 48.1               | 66.6              | 33.4               |
> | Qwen3-30-A3B   | LoPRo     | W3A16    | 96.07%     | 58.2           | 75.5         | TL               | 51.2          | TL            | 55.1               | 70.5              | 38.8               |
> | Qwen3-30-A3B   | LoPRo_v   | W2A16    | 87.76%     | 53.8           | 65.8         | TL               | 45.5          | TL            | 51.6               | 67.8              | 35.2               |
> | Qwen3-30-A3B   | LoPRo_v   | W3A16    | 95.38%     | 57.8           | 75.2         | TL               | 51.8          | TL            | 54.4               | 69.3              | 38.3               |
>
>
>
> - For the 235B model, a full evaluation on the Open LLM Leaderboard would incur prohibitive computational costs (exceeding 3,000 GPU hours), which is beyond our current resource capacity. Therefore, we only performed perplexity evaluations on the WikiText-2 and C4 datasets; the results are also included in the table below.
> | Models               | Methods   | Q Config | Wiki  | C4    |
> |----------------------|-----------|----------|-------|-------|
> | Qwen3-235b-A22b      | Fp16      | W16A16   | 5.37  | 8.59  |
> | Qwen3-235b-A22b      | LoPRo     | W2A16    | 6.84  | 11.21 |
> | Qwen3-235b-A22b      | LoPRo     | W3A16    | 5.56  | 9.12  |
> | Qwen3-235b-A22b      | LoPRo_v   | W2A16    | 6.31  | 10.13 |
> | Qwen3-235b-A22b      | LoPRo_v   | W3A16    | 5.55  | 9.16  |

---

### Note · Authors · 2026-01-28

I have read and agree with the venue's withdrawal policy on behalf of myself and my co-authors.

---

### Meta-Review · Area_Chair_TnzC · 2026-01-08

**Summary:**

Reviewers acknowledge solid empirical results but note that rotation-based quantization is well-established (SmoothRot), missing baselines (QTIP), and limited model diversity. While the paper adds evidence to the literature, it represents an incremental contribution to an already crowded area. Given the borderline scores and incremental nature, I recommend rejection.

**Reviewer Concerns:**

see above

**Reviewer Scores:**

Discussion was sufficient; reviewers largely agree on the incremental nature of the contribution, and scores would have remained similar.

---

### Decision · Program_Chairs · 2026-01-26

Reject